# Gene Therapy with Enterovirus 3 C Protease: A Promising Strategy for Various Solid Tumors

Xiaotong Yang[1,2,5], Wei Li[1,5], Shaokang Yang[1,5], Zhuang Wang[1], Jiye Yin[3], Wenhao Zhang[3], Huimin Tao[1,2], Siqi Li[1], Xiaojia Guo[1], Qingsong Dai[1], Weiyan Zhu[1], Yuexiang Li[1], Xintong Yan[1,4], Chongda Luo[1,4], Jiazheng Li[1,4], Sichen Ren[1,4], Ping Wang[1,4], Yunfeng Shao[1,4], Yan Luo[1,4], Zhenyang Li[1], Jingjing Yang[1,4], Zhijie Chang ☉ [2], Ruiyuan Cao[1] ✉, Song Li[1,4] ✉ & Wu Zhong ☉ [1] ✉

Current cancer gene therapies rely primarily on antitumor immunity, but the exploration of alternative mRNA cargoes for direct antitumor effects is crucial to expand cancer gene therapies. Here we show that lipid nanoparticles (LNPs) carrying mRNA encoding a viral 3 C protease can efficiently suppress tumors by selectively inducing tumor cell apoptosis. In various solid tumor models, intracranial injection of LNPs carrying mRNA encoding the 3 C protease (3C-LNPs) significantly inhibits tumor growth and prolongs survival in glioblastoma models. Similarly, subcutaneous injection reduces tumor volume and inhibits angiogenesis in a breast cancer model, while intravenous injection inhibits tumor growth and angiogenesis and prolongs survival in hepatocellular carcinoma models. Mass spectrometry and cleavage site prediction assays identify heterogeneous nuclear ribonucleoprotein A1 (hnRNP A1) as the main target degraded by the 3 C protease. This study suggests that viral protease mRNA could be a promising broad-spectrum antitumor therapeutic.

mRNA-based cancer therapies have emerged as promising approaches in the field of oncology, leveraging the ability to encode therapeutic proteins directly within the tumor microenvironment[1]. Recent advancements have demonstrated the potential of mRNA to express tumor antigens, immunomodulatory proteins, and other therapeutic agents, offering a versatile platform for cancer treatment[2]. However, the clinical application of mRNA therapies faces several challenges, including the need for efficient and targeted delivery, the stability of mRNA molecules, and the potential for immune activation[1]. Lipid nanoparticles (LNPs) have become a leading delivery system for mRNAs because of their ability to protect mRNAs from degradation and facilitate their cellular uptake[2–4].

At present, tumor-targeting LNPs carrying various cargoes, such as tumor antigens, CRISPR-Cas9, chimeric antigen receptors (CARs), T-cell receptors (TCRs), and cytokine mRNAs, have been reported[5–10]. However, the adaptability of these approaches for treating immuno-compromised patients is unclear and additional issues, such as the off-target effects of Cas9, the economic and time constraints of T-cell extraction and the manufacturing process, and the potential to generate cytokine storms, are of concern to researchers. Therefore, exploring alternative mRNA cargoes to achieve direct and selective tumoricidal effects is critical for expanding mRNA-LNP antitumor approaches.

Inspired by targeted protein degradation (TPD) strategies including PROTAC, LYTAC and molecular glues, we use a viral protease

[1]National Engineering Research Center for the Emergency Drug, Beijing Institute of Pharmacology and Toxicology, Beijing, China. [2]School of Basic Medical Sciences, Tsinghua University, Beijing, China. [3]National Beijing Center for Drug Safety Evaluation and Research, Beijing Institute of Pharmacology and Toxicology, Beijing, China. [4]Song Li's Academician Workstation of Hainan University (School of Pharmaceutical Sciences), Yazhou Bay, Sanya, China. [5]These authors contributed equally: Xiaotong Yang, Wei Li, Shaokang Yang. ✉e-mail: 21cc@163.com; lis.lisong@gmail.com; zhongwu@bmi.ac.cn

to achieve direct and selective gene degradation[11]. Enterovirus (EV) is a self-limiting human virus that mainly infects infants and children, and the EV-3C protease is able to specifically cleave proteins at glutamine/glycine (*Q/G*) and glutamine/serine (*Q/S*) sequences[12–14]. Considering self-limiting infections such as those caused by enteroviruses and coxsackieviruses, as well as the role of the viral protease as a virulence factor, we hypothesize that the use of proteases from these sources may be exploited as an mRNA-based antitumor approach.

In this work, we demonstrate the potential of LNPs carrying mRNA encoding a viral 3 C protease as a versatile antitumor strategy. Our results show that 3C-LNPs can be effectively delivered via multiple routes (intracranial, subcutaneous, and intravenous) to treat various solid tumor models, including glioblastoma, breast cancer, and hepatocellular carcinoma. This strategy safely and effectively inhibits tumor growth in vivo while prolonging the survival of tumor-bearing mice. Through mass spectrometry and cleavage site prediction assays, we identify hnRNP A1 as a primary target of the 3 C protease, underscoring the therapeutic potential of our mRNA-based approach. Our findings indicate that LNPs harboring a single hetero(viral)-protease mRNA may serve as a potential strategy for tumor treatment.

## Results

### 3C-mRNA has antitumor activity in vitro

Glioblastoma (GBM) is a highly aggressive and malignant tumor, and despite continuous improvements in the treatment of GBM, the 5-year survival rate of patients is still less than 10%[15,16]. Given the refractory nature of GBM, we attempted to evaluate the antitumor activity of enterovirus 3 C protease mRNA toward GBM cells. The 3C-mRNAs synthesized included the codon-optimized 3 C region of the human enterovirus, the 5' untranslated region (5'UTR), the 3' untranslated region (3'UTR), and a poly (A) tail (Fig. 1a). Following 3C-mRNA transfection, the 3 C protein was successfully expressed in both U87 MG and U138 MG cells (Fig. 1b). To evaluate whether 3C-mRNA has a tumor suppressive effect, GBM cells were subsequently transfected with different amounts of 3C-mRNA, and their viability evaluated with the use of adenosine triphosphate (ATP), an indicator of living cell metabolism. We found that the viability of U87 MG and U138 MG cells decreased with increasing amounts of 3C-mRNA (Fig. 1c). To validate whether the antitumor activity of 3C-mRNA depends on its protease activity, the chemical 3C^pro inhibitor AG7088[17,18] and the 3C^pro C147S mutant[19], which completely lacks protease activity, were used. As shown in Fig. 1d, AG7088 dose-dependently ameliorated the 3C-mRNA-induced death of GBM cells. Moreover, 3 C(C147S)-mRNA did not reduce the viability of GBM cells (Fig. 1e).

The proliferation of 3C-mRNA-treated cells was evaluated using an EdU staining assay, and the results suggested that, compared with that of the control group, the proliferation of cells transfected with 3C-mRNA was significantly inhibited in a dose-dependent manner. In contrast, no significant inhibitory effect on cell growth was observed in cells transfected with 3 C(C147S)-mRNA or 3C-mRNA and treated with AG7088 (Fig. 1f). Collectively, these results indicate that the antitumor effect of 3C-mRNA is dependent on its protease activity.

Given that 3 C exhibited potent tumor-suppressive effects on GBM cells, we next evaluated the effects of 3C-mRNA in various tumor cell lines and normal cells (Supplementary Fig. 1 and Supplementary Table 1). The results showed that 3C-mRNA had no significant inhibitory effect on normal cells (HUVECs), even those transfected with up to 10000 ng/mL 3C-mRNA. Several of the tested tumor cell lines exhibited significantly decreased cell viability with increasing 3C-mRNA concentrations in vitro. Taken together, these results indicate that 3C-mRNA has broad-spectrum antitumor activity in vitro and is almost nontoxic to normal cells.

### 3C-mRNA suppresses proliferation, migration, and invasion, and increases apoptosis in GBM cell lines

Given the antitumor activity of 3C-mRNA, we next sought to assess the tumor-suppressing properties of 3C-mRNA by investigating its effects on migration, invasion, and apoptosis in GBM cell lines. U87 MG and U138 MG cells were transfected with 3C-mRNA, 3 C(C147S)-mRNA and AG7088. The results of the Transwell migration and invasion assays indicated that transfection with 3C-mRNA suppressed the mobility and invasiveness of GBM cells (Fig. 2a, b). Conversely, transfection with 3 C(C147S)-mRNA or 3C-mRNA combined with AG7088 treatment had no significant inhibitory effect on cell migration or invasion.

The effects of 3C-mRNA, 3 C(C147S)-mRNA and 3C-mRNA combined with AG7088 on GBM cell apoptosis were subsequently evaluated in U87 MG and U138 MG cells using flow cytometry. As shown in Fig. 2c, d, the proportion of apoptotic cells among cells transfected with different amounts of 3C-mRNA increased significantly, with the maximal increase in apoptosis observed in cells transfected with at 1.5 μg/mL 3C-mRNA. Taken together, these findings revealed a decrease in proliferation, migration, and invasion and an increase in apoptosis following 3C-mRNA transfection, which indicates that 3C-mRNA has tumor-suppressor effects in GBM.

### 3C-LNPs suppress tumor development and growth in orthotopic GBM models

Having demonstrated its tumor suppression efficacy in vitro, we next focused on the vehicle for delivering 3C-mRNA in vivo. We chose lipid nanoparticles (LNPs), which have been shown to be safe and effective in delivering the Moderna COVID-19 mRNA vaccine (Supplementary Fig. 2). The results revealed that the mRNA encapsulation efficiency of the 3C-LNPs was 94.41%, and the average particle size was 82.76 nm.

The tumor suppression potential of 3C-LNPs in vivo was then investigated in BALB/c nude mouse xenograft models of GBM established by planting U87 MG-luc cells in the mouse brain. Seven days after tumor cell implantation, the mice were randomly divided into four groups. Two dosing strategies were used to administer 3C-LNP: 1.2 μg once per week or 1.2 μg twice per week for three weeks. PBS and 3 C(C147S)-LNPs were administered to the control groups (Fig. 3a). The bioluminescence signals in the tumors of the mice were monitored. In the PBS and 3 C(C147S)-LNP groups, an increase in bioluminescence was observed from the first week (W1) to the fifth week (W5), indicating continued tumor growth. However, compared with that in the PBS and 3 C(C147S)-LNP groups, the bioluminescence signal observed in the 3C-LNP treatment group gradually decreased. This decrease was especially evident in the 3C-LNP (1.2 μg twice per week) group, in which bioluminescence signals were not detected in three mice on Day 28 (Fig. 3b, c). In addition, compared with PBS, 3C-LNPs significantly protected the mice from death, prolonged survival, and suppressed tumor growth (Fig. 3d–f).

Using the same model, we also examined the effect of 3C-LNPs on intratumoral angiogenesis (Fig. 3g). As shown in Fig. 3h, the tumor blood vessel area density and vessel skeleton density were significantly lower in the 3C-LNP-treated group than in the PBS-treated group. Collectively, these results indicate that 3C-LNPs can effectively inhibit the development and growth of GBM and significantly prolong survival in tumor-bearing mice.

### The antitumor activity of 3C-LNPs is independent of the tumor immune response

To elucidate the correlation between the antitumor activity of 3C-LNPs and antitumor immunity, we established two synergistic models. Specifically, GL261 cells were used to construct a subcutaneous tumor model in C57L/6 mice. 4T1 cells were utilized to establish a mammary fat pad tumor model in BALB/c mice (Supplementary Fig. 3). Four days after the inoculation of GL261 cells, the tumor-bearing mice were randomly divided into control and treatment groups, which were

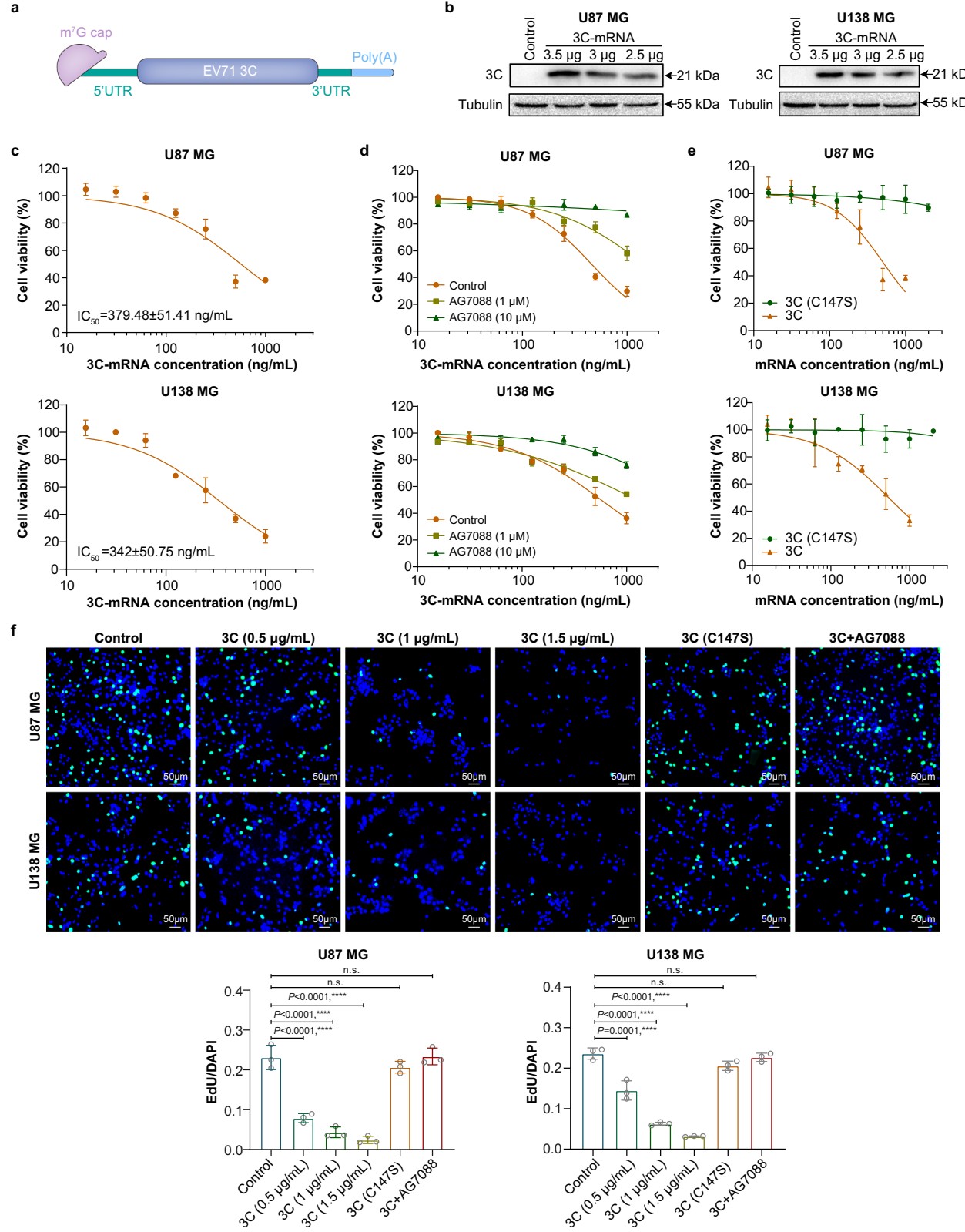

intratumorally administered PBS or 3C-LNPs (3 µg), respectively, every 3 days for a total of four doses (Supplementary Fig. 3a). Compared with the mice in the PBS-treated group, the mice in the 3C-LNP-treated group exhibited reduced tumor growth, as indicated by smaller tumor volumes (Supplementary Fig. 3b). To assess whether the administration of 3C-LNPs induced the tumor immune microenvironment, we collected tumor tissues 24 hours after the first administration for immunohistochemistry (IHC) and flow cytometric analysis. IHC staining revealed no significant difference in the proportion of CD8-positive or CD4-positive T cells in tumor tissues (Supplementary Fig. 3c–e). In addition, the quantification of cytotoxic T lymphocytes and regulatory T lymphocytes revealed no difference between the numbers of these cells in the PBS-treated and 3C-LNP-treated tumors (Supplementary Fig. 3f, g). Accordingly, syngeneic mammary tumor

**Fig. 1 | 3C-mRNA inhibits the viability and proliferation of GBM cells.**
**a** Schematic showing the mRNA construct containing the EV71 3 C transcript for expression in cells following transfection. The mRNA begins with an $m^7G$ cap, indicated in pink. Following the $m^7G$ cap is the 5'UTR, shown in green. The central and most substantial part of the molecule is the EV71 3 C coding sequence, depicted in purple. This region is flanked by the 3'UTR, shown in green, similar to the 5'UTR. The 3' end of the mRNA is the poly(A) tail, shown in light blue. **b** Western blot analysis of the 3 C protein in U87 MG and U138 MG cells transfected with 3C-mRNAs (2.5 μg, 3 μg, or 3.5 μg). **c** Cell viability assay of U87 MG and U138 MG cells transfected with different concentrations of 3C-mRNA for 24 h. **d** Cell viability assay of

cells transfected with different concentrations of 3C-mRNA and grown in the presence of AG7088 (10 μM, 1 μM, or 0 μM) for 24 h. **e** Cell viability assay of cells transfected with different concentrations of 3C-mRNA or 3 C(C147S)-mRNA for 24 h. **f** Cell proliferation ability of U87 MG and U138 MG cells treated with different concentrations of 3C-mRNA, 3 C(C147S)-mRNA (1.5 μg/ml), or 3C-mRNA+AG7088. Scale bars, 50 μm. **c–f** The data are presented as the means ± SDs. *n* = 3 independent biological samples. Statistical differences in pane (**f**) were assessed using one-way ANOVA with the Bonferroni multiple comparisons test. ****$P < 0.0001$, n.s.= not significant. Source data are provided as a Source Data file.

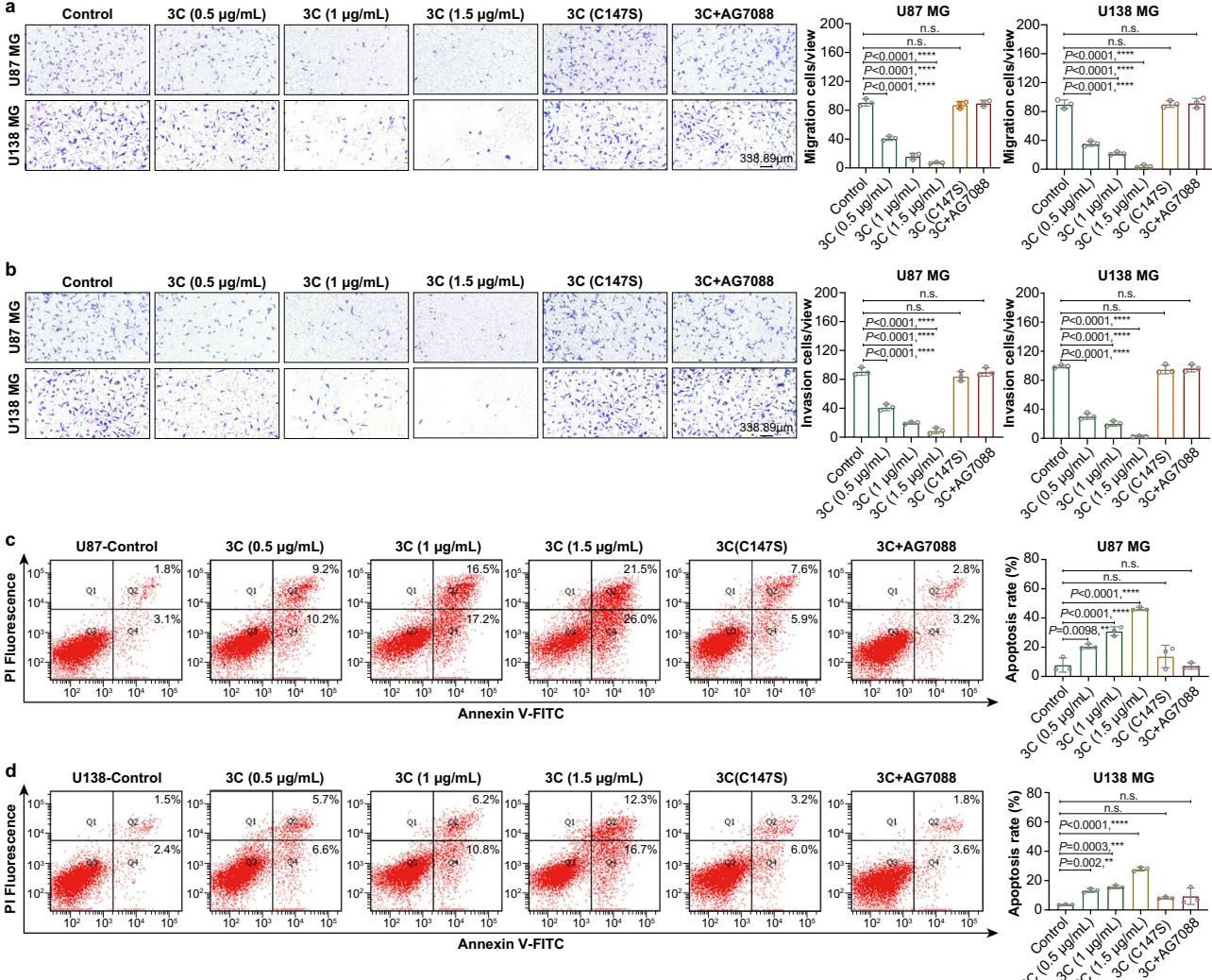

**Fig. 2 | 3C-mRNA inhibits the proliferation, migration and invasion of GBM cells and promotes cell apoptosis.** U87 MG and U138 MG cells were transfected with 3C-mRNA (0.5 μg/mL, 1 μg/mL, or 1.5 μg/mL), 3 C(C147S)-mRNA (1.5 μg/mL), or 3C-mRNA (1.5 μg/mL) + AG7088 (10 μM). **a** Cell migration ability of U87 MG and U138 MG cells subjected to different treatments, as determined by an in vitro Transwell migration assay. **b** Cell invasion ability of U87 MG and U138 MG cells subjected to different treatments, as determined by an in vitro Transwell invasion assay. Scale

bar: 338.89 μm. **c**, **d** Apoptosis was assessed 24 h after transfection with different concentrations of 3C-mRNA, 3 C(C147S)-mRNA, or 3C-mRNA+AG7088. The data are presented as the means ± SDs. *n* = 3 independent biological samples. Statistical differences were assessed using one-way ANOVA with the Dunnett multiple comparisons test. **$P < 0.01$, ***$P < 0.001$, ****$P < 0.0001$, n.s. = not significant. Source data are provided as a Source Data file.

models were further utilized to explore the immune response (Supplementary Fig. 3h). The results revealed that 3C-LNPs could also inhibit the growth of syngeneic mouse tumors (Supplementary Fig. 3i). The immunohistochemical results shown in Supplementary Fig. 3j–l, along with the flow cytometry analysis (Supplementary Fig. 3m, n), suggested that there was no significant difference between the control group and the 3C-LNP treatment group.

Furthermore, to comprehensively assess the contribution of the immune system to the therapeutic efficacy of the 3C-LNP strategy, we performed a series of T, B, and NK cell depletion experiments in C57BL/6 mice (Supplementary Fig. 4). Despite the depletion of CD3[+] T cells, CD19[+] B cells, and NK1.1[+] NK cells, the 3C-LNP treatment group still exhibited pronounced inhibition of tumor growth relative to the PBS control group.

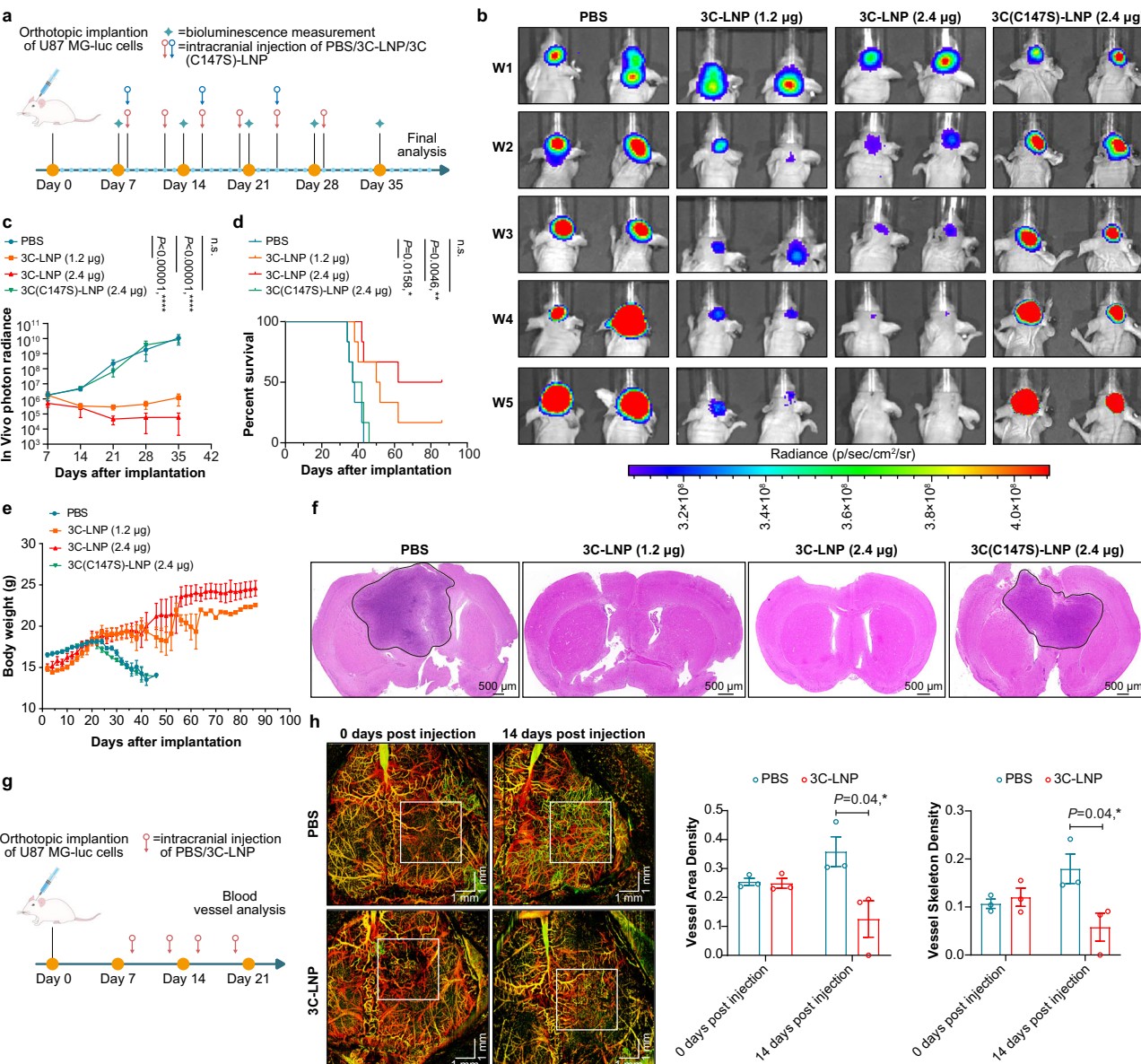

**Fig. 3 | 3C-LNPs suppress tumor growth and development in an orthotopic glioblastoma model. a** Timeline of tumor implantation and treatment schedule in the intracranial glioblastoma model. Mice with orthotopic U87 MG-luc tumors were intracranially administered 3C-LNPs (1.2 μg/week or 2.4 μg/week), 3 C(C147S)-LNPs (2.4 μg/week) or an equal volume of PBS. **b** and **c** Bioluminescence images of luciferase-expressing U87 MG-luc tumors obtained after 1, 2, 3, 4 and 5 weeks of treatment (*n* = 6 mice/group). The data are presented as the means ± SEMs. Statistical differences were assessed via mixed effects with the Dunnett multiple comparisons test. \*\*\*\**P* < 0.0001, n.s.= not significant. **d** Survival was assessed in the U87 MG-luc orthotopic mouse model using the log-rank (Mantel−Cox) test (*n* = 6 mice/group). **e** Body weight changes in tumor-bearing mice (*n* = 6 mice/group). **f** Representative H&E staining of tumor tissues from tumor-bearing mice from 6 mice/group. Scale bar, 500 μm. **g** Timeline of the tumor vascular microcirculation monitoring test in orthotopic U87 MG-luc tumor-bearing mice treated with 3C-LNPs (2.4 μg/week for two weeks) or PBS. **h** Representative images of the functional vasculature of 3C-LNP-treated or PBS-treated tumors (*n* = 3 mice/group). Scale bar: 1 mm. The data are presented as the means ± SEMs. Statistical differences were assessed using a two-tailed unpaired *t* test. \**P* < 0.05, \*\**P* < 0.01, \*\*\*\**P* < 0.0001, n.s.= not significant. Source data are provided as a Source Data file.

Collectively, these findings indicate that the therapeutic impact of 3C-LNPs may be mediated through mechanisms that are independent of adaptive and innate immune responses.

## 3C-LNPs suppress tumor development and growth in orthotopic breast cancer models

Given the broad antitumor activity of 3C-mRNA in vitro, we next sought to evaluate the therapeutic efficacy of 3C-LNPs via different administration routes. Orthotopic breast tumor models were established by implanting MCF-7 cells into the mammary fat pads of BALB/c nude mice. Four days after implantation, the nude mice were randomly divided into three groups, which were s.c. administered PBS, 3C-LNPs (1.5 μg), or 3C-LNPs (3 μg) around the tumor tissues every 3 days for a total of four doses (Fig. 4a and Supplementary Fig. 5). As shown in Fig. 4b–d and Supplementary Fig. 5, the tumors of the mice that were administered 3C-LNPs were smaller than those of the mice that were administered PBS, and the volume of tumors from the mice in the 3C-LNP (3 μg) treatment group was markedly lower than that of the mice in the other groups.

To assess the effect of 3C-LNPs on intratumor angiogenesis in this breast cancer model, an in vivo vascular microcirculation imaging assay was employed (Fig. 4e, f). The results revealed that the tumor

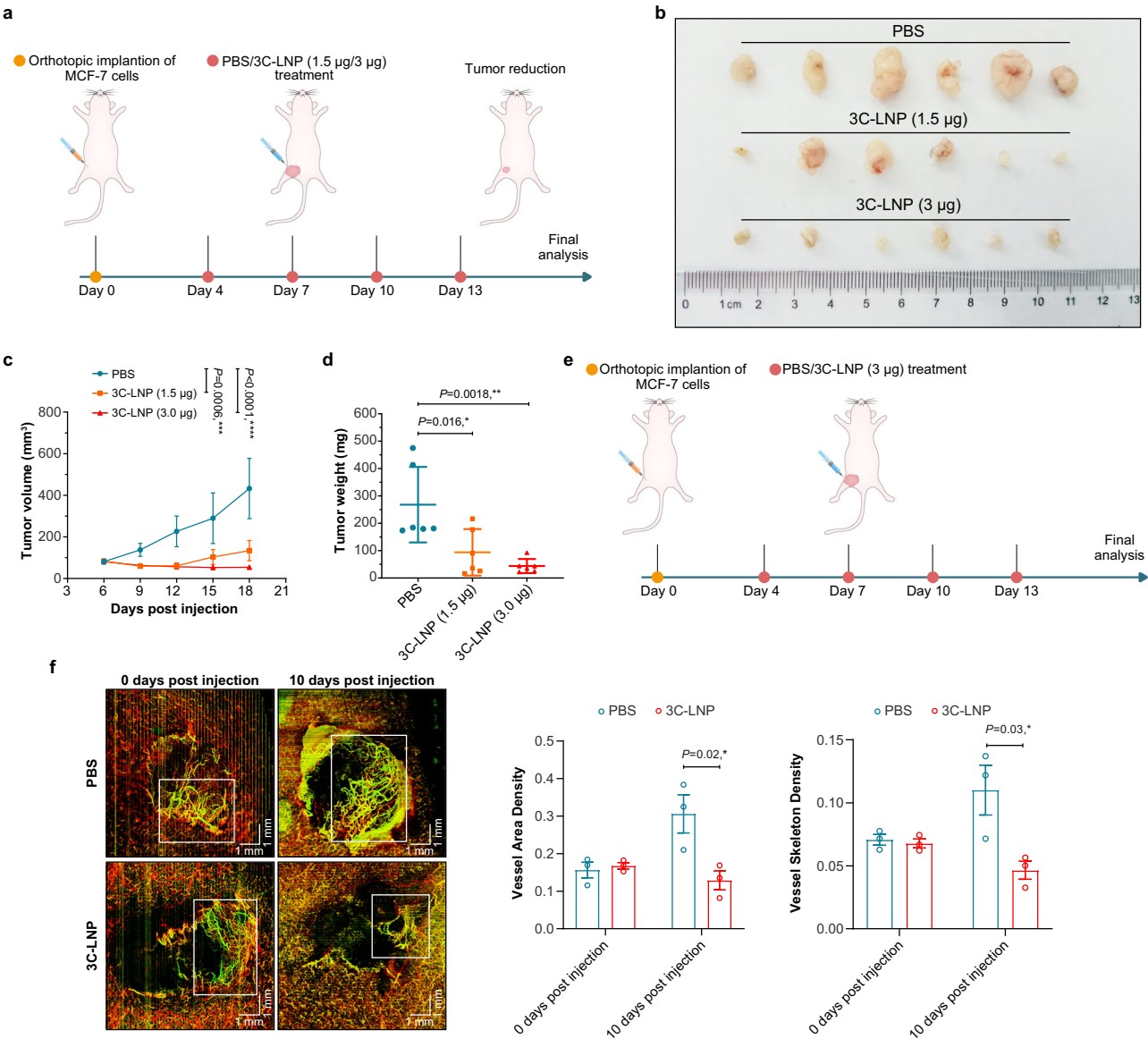

**Fig. 4 | Tumor suppression by 3C-LNPs in a breast tumor model. a** Timeline of tumor implantation and treatment schedule in the orthotopic breast tumor model. Mice were subcutaneously injected around the tumor tissues with PBS, 3C-LNPs (1.5 µg), or 3C-LNPs (3 µg) every 3 days for a total of four doses. **b** Representative images of tumors after treatment with PBS or 3C-LNPs ($n = 6$ mice/group). **c** Growth curves of tumors from tumor-bearing mice after treatment with PBS or 3C-LNPs ($n = 6$ mice/group). The data are presented as the means ± SEMs. Statistical differences were assessed using Mixed-effects with the Dunnett multiple comparisons test. ***$P < 0.001$, ****$P < 0.0001$. **d** Changes in tumor weight in tumor-bearing mice after the indicated treatments ($n = 6$ mice/group). The data are presented as the means ± SEMs. Statistical differences were assessed using one-way ANOVA with the Bonferroni multiple comparisons test. *$P < 0.05$, **$P < 0.01$. **e** Schematic showing the timeline of the tumor vascular microcirculation monitoring test. The tumor-bearing mice were treated with 3C-LNPs (3 µg every 3 days for four doses) and PBS. **f** Representative images of the functional vasculature of tumors from mice administered 3C-LNPs and from those administered PBS ($n = 3$ mice/group). Scale bar: 1 mm. The data are presented as the means ± SEMs. Statistical differences were assessed using a two-tailed unpaired $t$ test. *$P < 0.05$, **$P < 0.01$, ***$P < 0.001$, ****$P < 0.0001$. Source data are provided as a Source Data file.

blood vessel area density and the skeleton density were significantly reduced in the 3C-LNP treatment group. Overall, 3C-LNPs inhibited breast tumor growth and angiogenesis in vivo, suggesting that 3C-LNPs could be useful for breast cancer therapy.

## 3C-LNPs suppress tumor development and growth in orthotopic hepatocellular carcinoma models

Given the liver-targeting properties of the LNP formula used, the therapeutic efficacy of intravenous administration was evaluated in an orthotopic hepatocellular carcinoma (HCC) model. We used luciferase-expressing Huh7 cells to construct an orthotopic HCC

model in which tumor size can be assessed through bioluminescence monitoring. Seven days after the implantation of the cells, the mice were randomly divided into two groups and i.v. administered either PBS or 3C-LNPs (0.4 mg/kg) every 7 days for a total of 3 doses (Fig. 5a). In vivo imaging (Fig. 5b), in vivo photon radiance (Fig. 5c), and histopathological (H&E) staining (Fig. 5f) revealed that, compared with PBS, administration of 3C-LNPs significantly inhibited tumor growth. Moreover, i.v. administration of 3C-LNPs prolonged the survival of mice with orthotopic HCC tumors (Fig. 5d). We also performed an in vivo vascular microcirculation monitoring assay, the results of which showed that 3C-LNPs significantly reduced the area density and

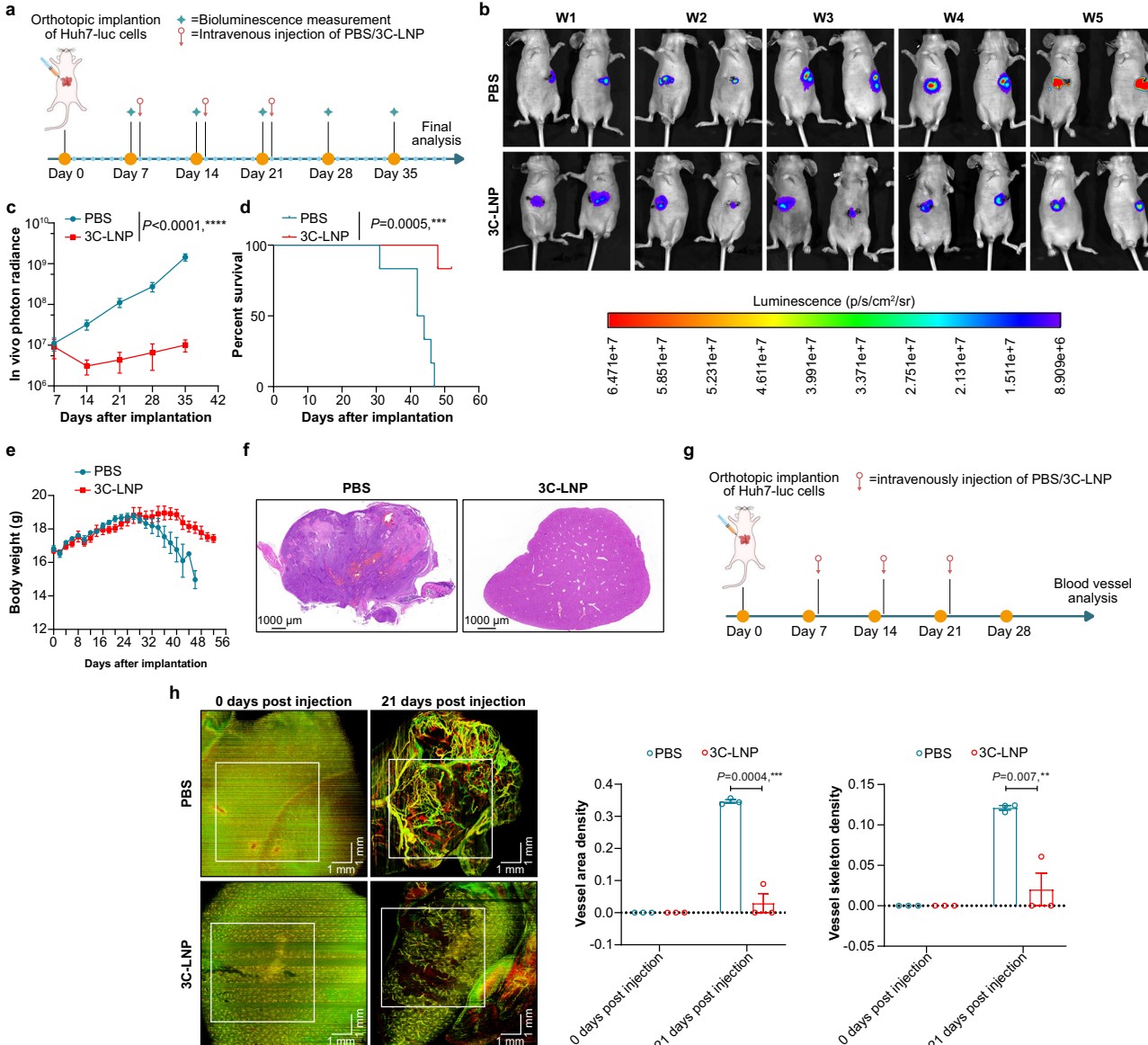

**Fig. 5 | Therapeutic efficacy of 3C-LNPs in an orthotopic HCC model. a** Timeline of tumor implantation and treatment schedule in the orthotopic (HCC) model. Mice were intravenously injected with PBS or 3C-LNPs (0.4 mg/kg) once every 7 days for a total of 3 doses. **b** and **c** Bioluminescence images of luciferase-expressing Huh7-luc tumors after 1, 2, 3, 4 and 5 weeks of treatment (*n* = 6 mice/group). The data are presented as the means ± SEMs. Statistical differences were assessed using Mixed-effects with the Bonferroni multiple comparisons test. ****P* < 0.0001. **d** Survival curve of the Huh7-luc orthotopic mouse model. Survival analysis was assessed using the log-rank (Mantel–Cox) test. ***P* < 0.001. **e** Changes in the body weights of the tumor-bearing mice in different groups (*n* = 6 mice/group). **f** Representative H&E staining of tumor tissues from 6 tumor-bearing mice/group. Scale bar: 1000 μm. **g** Schematic showing the timeline of the tumor vascular microcirculation monitoring test. The tumor-bearing mice were treated with 3C-LNPs (0.4 mg/kg) and PBS. **h** Representative images of the functional vasculature of tumors from mice administered 3C-LNPs or PBS (*n* = 3 mice/group). Scale bar: 1 mm. The data are presented as the means ± SEMs. Statistical differences were assessed using a two-tailed unpaired *t*-test. ***P* < 0.01, ****P* < 0.001. Source data are provided as a Source Data file.

skeleton density of tumor blood vessels (Fig. 5g, h). Taken together, these results confirm the therapeutic efficacy of i.v. administration of 3C-LNPs in HCC.

### Delivery of 3C-LNPs via different routes has a satisfactory safety profile

The potential toxicity and safety profile of an mRNA-LNP drug harboring a viral protease need to be systematically evaluated. To this end, BALB/c female nude mice were i.c. administered 1.2 μg 3C-LNPs in a 10 μL volume twice a week (2.4 μg/week) for 3 consecutive weeks (Fig. 6a). No significant differences in body weight curves and survival times were observed between the two groups (Fig. 6b, c).

Histopathological analysis of tissues from the 3C-LNP-treated and PBS-treated mice revealed no pathological changes (Fig. 6d). Next, we evaluated the safety of 3C-LNPs after i.v. administration to BALB/c mice injected with 3C-LNPs for 6 consecutive weeks (Fig. 6e). No significant differences in body weight curves or survival times were observed among the three groups (Fig. 6f, g). Toxicity was evaluated in these after 4 weeks of recovery, and no apparent clinical signs of toxicity were observed, as reflected by liver function (alanine transaminase, aspartate aminotransferase, and alkaline phosphatase) analyses (Fig. 6h) and total blood counts (Fig. 6i). Overall, these results indicated that 3C-LNPs administered via different dosing routes have an acceptable safety profile.

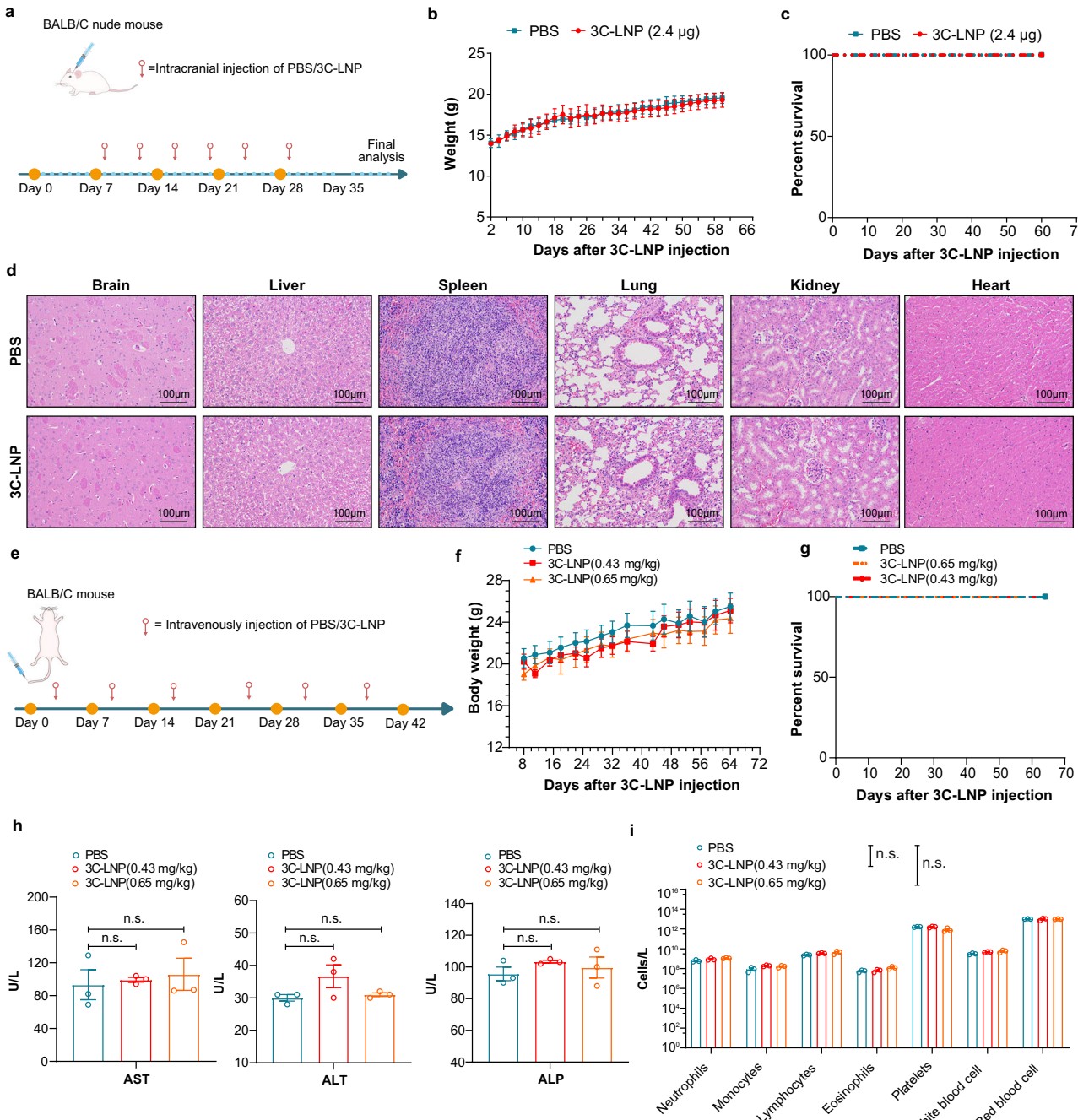

**Fig. 6 | Safety evaluation of 3C-LNPs. a–d** BALB/c female nude mice were intracranially administered 1.2 μg of 3C-LNPs twice per week for 3 weeks. **b** and **c** Body weight and survival curves of mice intracranially administered 3C-LNPs (*n* = 6 mice/group). **d** Histopathological examination of major organs collected after the animals were administered PBS or 3C-LNPs (*n* = 6 mice/group). Scale bar: 100 μm. **e–j** BALB/c mice were intravenously administered with 0.43 mg/kg or 0.65 mg/kg 3C-LNPs once a week for six weeks. **f, g** Body weight and survival curves of BALB/c mice (*n* = 6 mice/group). Liver function (**h**) (aspartate aminotransferase (AST),

alanine transaminase (ALT), and alkaline phosphatase (ALP)) and total blood cell counts. The data are presented as the means ± SEMs. Statistical differences were assessed using one-way ANOVA with the Dunnett multiple comparisons test. **i** were evaluated 4 weeks after six weeks of treatment (*n* = 3 mice/group). The data are presented as the means ± SEMs. Statistical differences were assessed using Multiple *t*-test with the Dunnett multiple comparisons test. n.s.=not significant. Source data are provided as a Source Data file.

## 3 C protease decreases the level of hnRNP A1 and induces apoptosis

To elucidate the antitumor mechanism of 3C-mRNA, coimmunoprecipitation (co-IP)-MS was used to identify proteins that interact with the 3 C protein in U87 MG cells. A total of 464 proteins and 281 differentially abundant proteins were identified on the basis of the quantified data (Supplementary Fig. 6). Among the 281 proteins, 100 proteins whose expression significantly differed were selected for

further analysis. Among them, 18 proteins were identified with multiple Q-G and Q-S amino acid sequences, which are the cleavage sites of 3 C proteases[12,14]. Among them, higher levels of major vault protein (*MVP*), disulfide-isomerase A6 (*PDIA6*), and *hnRNP A1* were detected in GBM tumor tissues than in normal tissues (Gene Expression Profiling Interactive Analysis database). We further investigated whether the 3C-mRNA-encoded 3 C protease affects the abundance of MVP, PDIA6, and hnRNP A1 proteins. Western blotting analysis revealed no

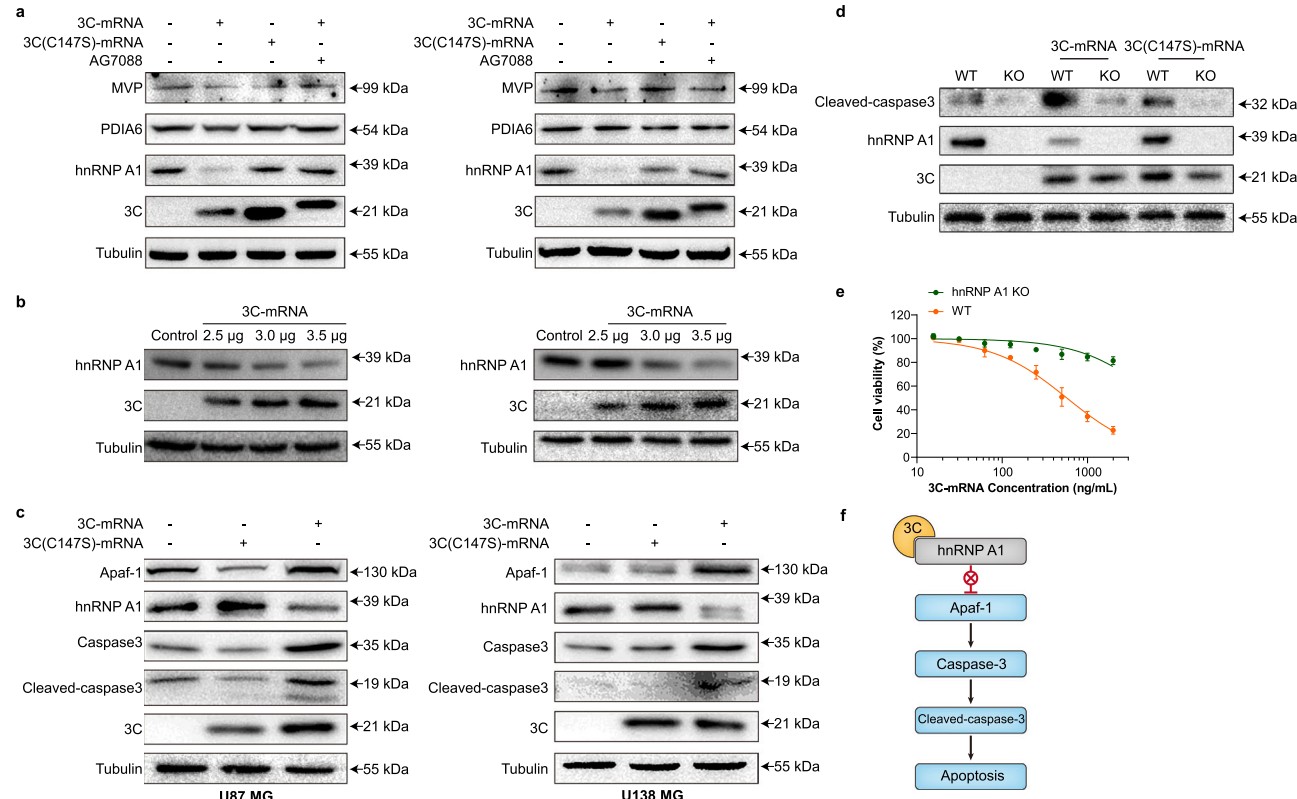

**Fig. 7 | Ectopic expression of 3 C protease decreases the level of hnRNP A1 and induces apoptosis through cleavage of hnRNP A1. a** Western blot analysis of MVP, PDIA6, hnRNP A1 and 3 C expression in U87 MG and U138 MG cells transfected with 3C-mRNA and 3 C(C147S)-mRNA and in cells transfected with 3C-mRNA and treated with AG7088. Representative blot images are shown. **b** The hnRNP A1 protein level decreased with increasing concentrations of 3C-mRNA. **c** Western blot analysis of hnRNPA1, Apaf1, Caspase-3, and cleaved-caspase-3 expression and 3 C expression in U87 MG and U138 MG cells transfected with 3C-mRNA and 3 C(C147S)-mRNA. **d** Western blot analysis of cleaved caspase-3, hnRNPA1, and 3 C expression in U87 MG-WT and U87 MG-*hnRNP A1* KO cells transfected with 3C-mRNA and 3 C(C147S)-mRNA. Panels (**a**–**d**) are from 3 independent experiments. **e** Cell viability assay of U87 MG-WT and U87 MG-*hnRNP A1* KO cells transfected with different concentrations of 3C-mRNA for 24 h. The data are presented as the means ± SDs. *n* = 3 independent biological samples. **f** The proposed 3 C protease–hnRNP A1–apaf-1–caspase-3–cleaved caspase-3-apoptosis axis. Source data are provided as a Source Data file.

significant differences in MVP or PDIA6 protein levels among the 3C-mRNA, 3 C(C147S)-mRNA, and 3C-mRNA plus AG7088 treatment groups, as depicted in Fig. 7a. However, a notably reduced level of hnRNP A1 protein was observed in cells treated with 3C-mRNA, implying that hnRNP A1 is a substrate for the 3 C protease. Interestingly, the 3 C band intensity was slightly elevated in the AG7088-treated group. This could be attributed to AG7088, a potent and irreversible inhibitor of the human rhinovirus (HRV) 3 C protease[20], which is known to form a covalent bond with the active site cysteine of the 3 C protease, thereby inactivating it. The irreversible nature of this inhibition suggests that AG7088 might induce a conformational change in the 3 C protease, possibly impacting its interaction with other cellular factors or its overall stability.

We also utilized the Human Protein Atlas (HPA) to analyze differences in protein expression between GBM tumors and normal tissues. Despite the disparity in sample sizes within the HPA database, which includes 10 normal samples and 100 tumor samples, we identified HNRNPH1, FUS, and HNRNPU as proteins that are highly expressed in GBM. To further validate these findings and assess the specificity of 3C-directed cleavage of these proteins, we conducted WB validation (Supplementary Fig. 7). The results from our WB analysis indicated that the levels of these proteins remained unchanged following treatment with 3C-mRNA, further suggesting the selectivity of 3 C degradation.

The results of the co-IP experiment verified that the 3 C protease can interact with hnRNPA1 (Supplementary Fig. 8). Moreover, the levels of hnRNPA1 progressively decreased with increasing expression

of 3 C (Fig. 7b), whereas the hnRNPA1 mRNA levels remained unchanged (Supplementary Fig. 9). Additionally, in vivo studies using orthotopic xenograft GBM models confirmed that the expression of 3 C protein reduces the level of hnRNPA1 and induces apoptosis in tumor cells (Supplementary Fig. 10). Previous studies have identified hnRNPA1 as a potential nuclear protein target that can be cleaved by the 3 C protease[19]. Notably, the binding of hnRNPA1 to the internal ribosome entry site of apoptotic peptidase activator 1 (apaf-1) is abolished through 3C-mediated hnRNPA1 cleavage[21]. Western blot analysis of U87 MG and U138 MG cells transfected with different amounts of 3C-mRNA revealed that the levels of hnRNPA1 decreased with increasing levels of 3 C, indicating that ectopic 3 C expression leads to hnRNPA1 degradation (Fig. 7c). Additionally, we utilized CRISPR technology to generate an *hnRNPA1* knockout (KO) U87 MG cell line. Western blot analysis revealed that the basal level of cleaved-caspase-3 in the *hnRNPA1* KO cell line was relatively low and did not decrease further upon 3 C expression (Fig. 7d). Moreover, the viability of the *hnRNPA1* KO cells was not affected by increasing concentrations of 3C-mRNA (Fig. 7e). Collectively, these results suggest that ectopic 3 C protease expression reduces hnRNPA1 levels and triggers apoptosis through the cleavage of hnRNPA1 (schematic in Fig. 7f).

In addition, we have broadened our investigation to include a diverse range of normal and cancer cell lines to ensure a thorough assessment of 3 C protease expression. WB analysis was conducted in HUVECs, 293 T cells, Huh7 cells, and MCF-7 cells to evaluate the expression levels of the 3 C protease (Supplementary Fig. 11). Given that in vivo studies confirmed the efficacy of 3C-LNPs against breast

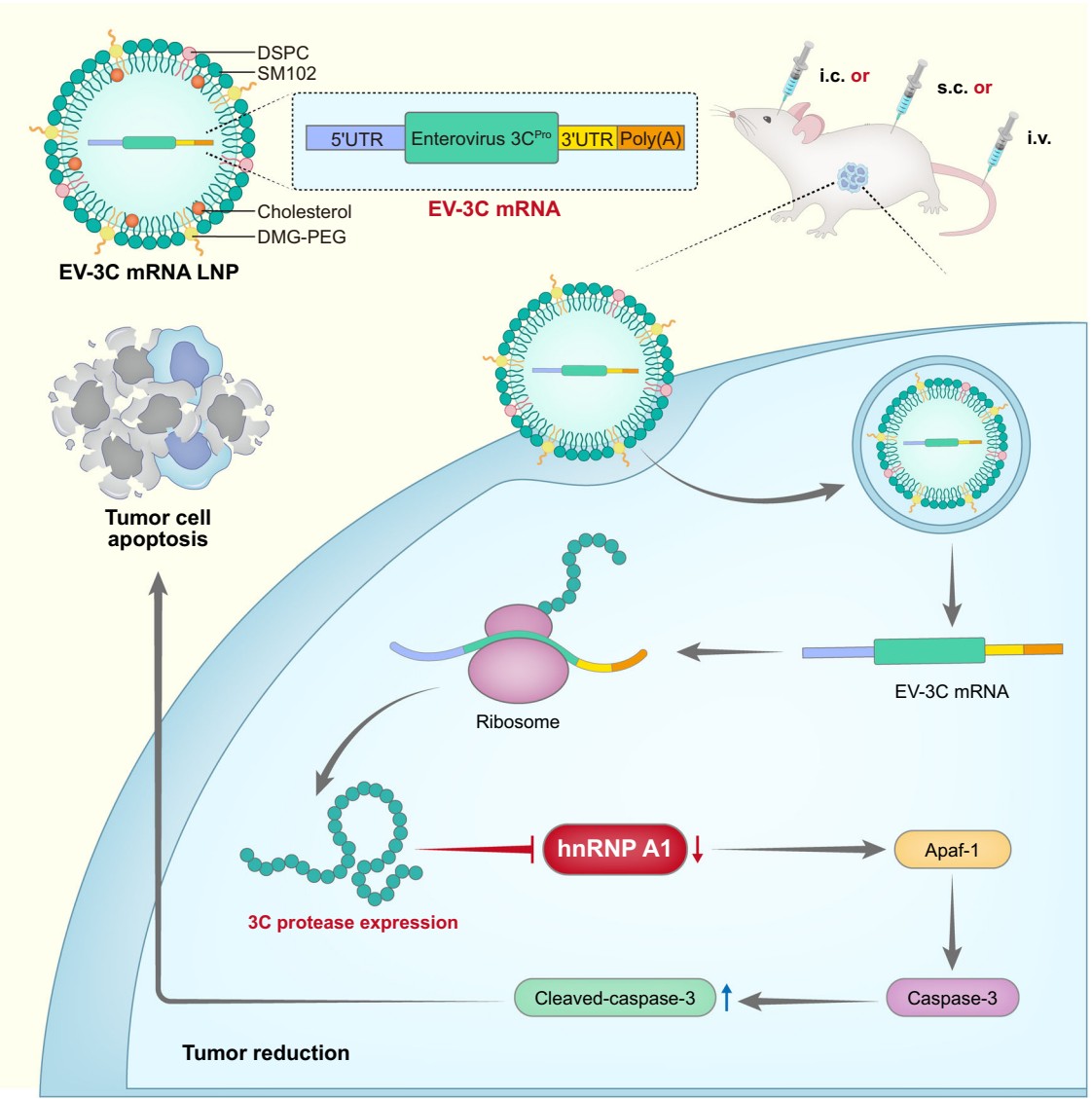

**Fig. 8 | Schematic illustration of apoptosis and tumor regression induced by EV 3C-mRNA.** The enterovirus 3 C protease mRNA was encapsulated into lipid nanoparticles to produce the 3C-LNPs, which efficiently induce tumor cell apoptosis by directly degrading hnRNP A1 and then activating caspase-3. Three dosing routes were tested in different mouse tumor models, each of which was shown to be highly efficacious in vivo. Intracranial (i.c.), subcutaneous (s.c.), or intravenous (i.v.) delivery routes were used for mouse tumor models of glioblastoma in the brain, breast cancer in the mammary gland, or hepatoma in the liver, respectively.

and liver cancers, we have conducted further in vitro validation. Additional WB analyses were performed to assess hnRNPA1 levels in Huh7 or MCF7 cell lines following 3C-mRNA treatment. The data revealed a significant reduction in hnRNPA1 levels upon 3C-mRNA treatment in both cell lines, which aligns with our findings in the GBM cell lines (Supplementary Fig. 12). Moreover, according to the HPA database, hnRNPA1 is indicated as a prognostic marker in liver cancer ($p < 0.001$), which underscores its relevance in HCC. (https://www.proteinatlas.org/ENSG00000135486-HNRNPA1/cancer).

The mouse cell lines used in our study included MEFs (mouse embryonic fibroblasts), GL261 cells (mouse glioma cells), and 4T1 cells (mouse breast cancer cells) (Supplementary Fig. 13). Our WB results demonstrated that 3 C protease expression is comparable among these mouse cell lines. Notably, the expression of 3 C led to a decrease in hnRNPA1 levels in these cell lines, suggesting that the 3 C protease is active in mouse cells and affects hnRNPA1, indicating a conserved mechanism between human and mouse cells. Collectively, our study elucidates the mechanism by which 3C-mRNA encapsulated in LNPs induces tumor cell apoptosis. These findings demonstrate that hnRNP

A1 is an important target of the 3 C protease, which subsequently leads to the activation of caspase-3 (Fig. 8).

## Discussion

Pathogen-derived heterologous proteases are key virulence factors that cleave specific host proteins and lead to pathological consequences, including autophagy, apoptosis, and pyroptosis[14,19,22–26]. Proteases cleave target proteins at conserved sites, and individual proteases often have a highly limited cleavage spectrum[12]. In our study, we developed an innovative mRNA-based antitumor strategy utilizing a heterologous (viral) protease. Specifically, human enterovirus 3 C protease mRNA was encapsulated into LNPs, which were subsequently administered to tumor-bearing mice and resulted in direct degradation of an antitumor target. The in vitro antitumor effects of 3C-mRNA, as well as their in vivo therapeutic effects of 3C-LNP were validated in multiple cell lines or tumor models, respectively. Interestingly, 3C-LNPs were highly effective and safe when administered via multiple dosing routes, including i.v., i.c., and s.c., in different tumor types.

Through a mechanistic study involving co-IP-MS and cleavage site prediction, hnRNP A1 was identified as a selective cleavage target of the 3 C protease. hnRNP A1 is the most studied member of the hnRNP A/B family and plays a key role in a variety of cancers[27]. hnRNP A1 is overexpressed in non-small cell lung cancer (NSCLC), and its level is negatively correlated with the overall survival in patients with lung cancer[28,29]. Additionally, hnRNP A1 is highly expressed in the sentinel lymph nodes, tissues, and serum of patients with hereditary colorectal cancer[30,31]. Furthermore, on the basis of existing research, hnRNP A1 is considered a potential cancer biomarker as well as a therapeutic target[27,32–34]. Despite efforts in the development of hnRNP A1 inhibitors such as the small-molecule inhibitor VPC-80051[35], the plant-extracted flavonoid quercetin[36], and the coumarin derivative esculetin[37] in anti-tumor studies, no drugs are currently approved against this drug target. Here we found that the 3 C protease interacted with hnRNP A1, reduced the protein level by cleaving hnRNP A1, and directly led to tumor cell apoptosis. Inspired by these results, other potential drug targets that were previously 'undruggable' may become 'druggable' by the hetero-protease mRNA-LNP approach.

Compared with current oncolytic virus approaches, the hetero-protease mRNA-LNP approach may eliminate potential immune storms and other risks associated with the use of live viruses. Notably, we demonstrated that the antitumor activity of 3C-LNPs was independent of the immune response, which has great advantages over therapies that rely on the body's immunity to achieve antitumor effects. The safety of the 3C-LNPs was confirmed by repeated i.v. and i.c. administration and the lack of side effects observed in safety assessments suggested that the protease did not cause negative off-target effects in normal tissues. Moreover, the off-target effects of this 3C-mRNA knockdown approach should be studied in further clinical trials.

This study provides proof-of-concept evidence that a single protein (such as a protease) could be developed as a drug. Considering that de novo functional protein design by machine learning could now be easily realized[38], we believe that our heteroprotease mRNA strategy might inspire the potential medicinal use for functional protein design in the future.

## Methods

### Ethics statement
This research complied with all relevant ethical regulations. All the experiments in this research were approved by the Institutional Animal Care and Use Committee (IACUC) of the Beijing Institute of Pharmacology and Toxicology (IACUC-DWZX-2022-622) (approved on 14 March 2022).

### In vitro transcription (IVT) of mRNAs
DNA templates for in vitro transcription were generated by inserting the EV71-3C (NP_042242.1) coding sequence between the 5' and 3' UTRs of the pmRVac vector. The pmRVac vector contains a T7 promoter sequence; a 5' UTR, a 3' UTR, a 110 nt segmented poly (A) sequence, and a SapI restriction enzyme site used for run-off transcription. The insertion site was located between the UTRs.

Cap1 mRNA was generated by the T7 in vitro transcription method and subsequent enzymatic capping and methylation. An in vitro transcription reaction was performed using a linear DNA template, T7 RNA polymerase, nucleotide triphosphates (NTPs) and magnesium-containing buffer to produce uncapped RNA transcripts in vitro. The reaction mixture was incubated for 2 h at 37 °C. For m1Psi-modified transcript synthesis, m1Psi was used in place of UTP for in vitro transcription. Cap1 mRNA was produced with a Vaccinia capping system. After denaturation of the uncapped transcripts by heating, the Vaccinia capping enzyme, 2'-O-methyltransferase, GTP, S-adenosyl methionine and capping buffer were added to begin the capping reaction. The reaction mixture was incubated for 1 h at 37 °C. The

template DNA was subsequently removed by DNase I treatment. The capped transcripts were further purified using magnetic beads.

The isolated mRNA was eluted with acidic buffer and stored at −80 °C. The mRNA concentration and purity were measured on a UV–Vis spectrophotometer. The mRNA integrity was evaluated by denaturing agarose gel electrophoresis and subsequent analysis.

### Lipid nanoparticles
Lipids (the ionizable cationic lipid SM-102, the helper lipid DSPC, cholesterol and the PEGylated lipid PEG2000-DMG) were dissolved in ethanol at a molar ratio of 50:10:38.5:1.5. The mRNA was diluted in sodium citrate acidification buffer to the desired concentration. LNPs were produced by mixing lipids with mRNA using a microfluidic mixer. The LNPs were subsequently dialyzed against Tris-HCl buffer with sucrose in a dialysis cassette. After dialysis, the LNPs were passed through a 0.22 μm filter and stored at −80 °C. A RiboGreen assay was used to quantify the mRNA in the LNPs. The particle size and surface charge were determined by dynamic light scattering and zeta potential measurements with a Zetasizer device. Endotoxin levels were measured by a kinetic chromogenic TAL assay.

### Cell lines
The following cell lines were purchased from the American Type Culture Collection (ATCC): U87 MG (HTB-14), HepG2 (HB-8065), U2OS (HTB-96), CHL-1 (CRL-9446), DU145 (HTB-81), HCT-116 (CCL-247), Hela (CCL-2), ACHN (CRL-1611), AGS (CRL-1739), and A549 (CCL-185). Huh7 cells were purchased from the National Experimental Cell Resource Sharing Platform. U138 MG,MCF-7, GL261 cells, U87 MG-luc cells and Huh7-luc cells were obtained from Hunan Fenghui Biotechnology Co., Ltd. HUVEC (CTCC-0804-PC) were obtained from Zhejiang Meisen Cell Technology Co., Ltd. All cell lines were routine validated using the Short Tandem Repeat (STR) profiling cell authentication service. The mouse-derived cell lines were subjected to species identification. For primary HUVEC cells, identification was conducted using immuno-fluorescence. The cells were routinely tested for mycoplasma contamination using PCR-based methods and were found to be negative. All cell lines were cultured as monolayers in Dulbecco's modified Eagle's medium (DMEM; Gibco, cat no. 2367374) supplemented with 10% fetal bovine serum (FBS; Gibco, cat no. 1807622) and 1% penicillin/streptomycin (PS; Gibco, cat no. 2321152) at 37 °C in the presence of 5% $CO_2$.

### Mice, tumor models, and treatment
All animal experiments were performed in accordance with the guidelines on the use and care of laboratory animals for biomedical research published by the National Institutes of Health and approved by the Beijing Institute of Pharmacology and Toxicology (IACUC-DWZX-2022-622) (approved on 14 March 2022). The maximal tumor size was limited to 2 cm in length, and the maximal tumor burden was not exceeded in all the animal experiments. To generate the orthotopic GBM model, 6–8-week-old female BALB/c nude mice (purchased from Sbeifer Biotechnology Co., Ltd.) were anesthetized via the intraperitoneal administration of 1% pentobarbital sodium (0.1 mL/10 g body weight), and the top of the head skin was subsequently sterilized with 75% ethanol. A vertical incision was made before the midline of the skull, and the anterior fontanel was exposed. The inoculation site was 0 mm before the midpoint of the fontanelle, 3 mm to the right of the sagittal suture, and 3 mm deep. A $1 \times 10^5$ U87MG-luc cell suspension (5 μL) was slowly injected using a Hamilton injector with a 27-G needle at an injection speed of 0.5 μL/min. The needle was then slowly removed after 2 min, and the scalp was sutured and disinfected with iodine. The volume that could be administered to the brains of the mice was limited to 10 μL. Therefore, we administered 3C-LNPs to the brain using two strategies: once a week (1.2 μg/week) and twice a week (2.4 μg/week).

For the orthotopic breast cancer model, $1 \times 10^7$ viable MCF-7 cells suspended in a Matrigel:PBS mixture (1:1 v/v) were injected into the left breast pads of 6–8-week-old female BALB/c nude mice. The mice were subcutaneously injected around the tumor tissues with PBS, 3C-LNPs (1.5 μg), or 3C-LNPs (3 μg) every 3 days for a total of four doses. For the orthotopic hepatocellular carcinoma model, $1 \times 10^6$ viable Huh7-luc cells were injected into the livers of 6–8-week-old female BALB/c nude mice. The mice in each group were intravenously injected with PBS or 3C-LNPs (0.4 mg/kg) every 7 days for a total of 3 doses.

For the subcutaneous syngeneic GL261 tumor model, $5 \times 10^6$ viable GL261 cells suspended in 100 μL of a Matrigel:PBS mixture (1:1 v/v) were injected into the right flank of 7–8-week-old female C57BL/6 mice (purchased from Sbeifer Biotechnology Co., Ltd.). Tumor-bearing mice were subcutaneously administered PBS or 3C-LNPs (3 μg) every 3 days for a total of three doses.

All the mice had unlimited access to a sterile rodent diet (pellets) and reverse osmosis-purified water and were maintained on a 12:12 h light:dark cycle at 22–24 °C and 45–55% humidity.

## In vivo bioluminescence imaging
Luciferase signals were analyzed by in vivo bioluminescence imaging (BLI) with the Xenogen IVIS Spectrum Imaging System (Caliper Life Sciences). An aqueous solution of L-luciferin (10 μL/g; BD Biosciences) was injected intraperitoneally into live mice. The luciferase activity was quantified in live animals 10 min later with an exposure time of 1 min. For imaging of the intracranial glioma xenograft model, BLI was performed every 7 days after tumor inoculation. Regions of interest (ROIs) were quantified as average radiance (photons/ [s cm² sr]) represented as color-scaled images superimposed on grayscale photos of the mice in Living Image software (Caliper Life Sciences).

## In vivo optical coherence tomography angiography (OCTA) imaging
All microvascular images were acquired by the Monitoring System of Vascular Microcirculation In Vivo (Micro-VCC, OPTOPROBE, Beijing HealthOLight Technology Co., Ltd.). The spectral bandwidth was 100 nm, the central wavelength was 1310 nm, and the axial resolution was 16 μm. This system can provide a wide field of view (FOV) of approximately 8 mm (x-axis) × 8 mm (y-axis) × 3 mm (z-axis). The mice were anesthetized with 2% isoflurane using a gas vaporizer (RWD Company, Shenzhen, China) and fixed on a high-precision translation stage to acquire the optimal imaging positioning and reduce motion artifacts. A heating pad was used to maintain the body temperature of the mice during imaging. Images of the tumor microvasculature were subsequently captured at 0 days, 14 days (GBM model), 10 days (breast tumor model) and 21 days (HCC model) after the first injection of 3C-LNPs.

## Immune Depletion in Mice
For CD3+ T-cell depletion, anti-CD3 (BioXCell, BE0001-1) and corresponding control (BioXCell, BP0091) IgG antibodies were administered i.p. (100 μg/dose), starting 4 days before tumor implantation, and continuing every 7 days thereafter.

For CD19+ B-cell depletion, anti-CD19 (BioXCell, BE0150) and corresponding control (BioXCell, BE0089) IgG2a antibody were administered i.p. (250 μg/dose), starting 4 days before tumor implantation, and continuing every 7 days thereafter.

For NK1.1+ NK-cell depletion, anti-NK1.1 (BioXCell, BP0036) and corresponding control (BioXCell, BP0085) IgG2a antibody were administered i.p. (250 μg/dose), starting 4 days before tumor implantation, and continuing every 7 days thereafter.

## CellTiter-Glo™ Luminescent Cell Viability Assay
3C-mRNA was transfected into U87 MG and U138 MG cells at concentrations of 998.4 ng/mL, 499.2 ng/mL, 249.6 ng/mL, 124.8 ng/mL,

62.4 ng/mL, 31.2 ng/mL and 15.6 ng/mL. For the protease activity inhibition assay, cells were treated with 3 C(C147S)-mRNA or with different concentrations of 3C-mRNA and AG7088 (10 μM, 1 μM, or 0 μM). After 24 h of treatment, cell viability was determined with a Promega Cell-Titer-Glo™ Luminescent Cell Viability Assay Kit.

## EdU staining
U87 MG and U138 MG cells were seeded in 6-well plates at $5.0 \times 10^5$ cells/well and transfected with 3C-mRNA (0.5 μg/mL,1 μg/mL and 1.5 μg/mL) or 1.5 μg/mL 3 C(C147S)-mRNA with Lipofectamine® 2000 Transfection Reagent (Thermo Fisher Scientific, 11668-019). For the AG7088 treatment group, after transfection with 1.5 μg/mL 3C-mRNA, 10 μM AG7088 was added to the medium. The cells were incubated at 37 °C for 24 h. Then, preheated 2× EdU solution was added to medium containing an equal volume of transfected cells. The treated cells were incubated for 2 h, after which the cells were fixed and permeabilized, and a click reaction was performed. To determine the extent of cell proliferation, Hoechst 33342 was used for nuclear staining.

## Transwell assay
U87 MG and U138 MG cells transfected with different concentrations of 3C-mRNA were incubated overnight for 12 h. The cells were harvested in FBS-free medium and seeded ($1 \times 10^4$ to $2 \times 10^4$ cells/well) in the upper Transwell chamber. DMEM + 10% fetal bovine serum (0.6 mL) was added to the lower Transwell chamber as an attractant, and the chamber was incubated at 37 °C for 16 h. The cells were fixed with methanol for 15 min and then stained with 1% crystal violet for 20 min. The membrane was washed with PBS for a few seconds to remove excess dye. The membrane was completely dried, and the cells were counted under a microscope. Different views were randomly selected, and the average count was determined.

## Flow cytometry apoptosis assays
At 12 h after transfection with 3C-mRNA, the cell pellet was collected and washed twice with cold PBS. The cells were then stained with PI and Annexin V-FITC at room temperature for 15 minutes. A FACSCalibur flow cytometer (Beckman Coulter, Atlanta, Georgia, USA) was subsequently used to determine the proportion of apoptotic cells.

## Immunoblotting and antibodies
U87 MG and U138 MG cells transfected with different concentrations of 3C-mRNA were incubated overnight. The cells were washed three times with cold PBS, followed by the addition of 150 μL of RIPA lysis buffer (APPLYGEN, cat no. 1053) containing a protease and phosphatase inhibitor cocktail (Thermo Fisher Scientific, cat no. A32961). The proteins were separated on a 10% SDS–PAGE gel and then transferred onto a polyvinylidene fluoride (PVDF) membrane. The PVDF membranes were blocked with 5% skim milk for 1 h at room temperature and then incubated with primary antibody at 4 °C overnight. The membranes were washed in TBST (150 mM NaCl, 50 mM Tris-HCl at pH 7.4, and 0.1% Tween 20) three times and incubated with horseradish peroxidase (HRP)-conjugated goat anti-mouse IgG antibody (1:5000) at room temperature for 1 h. Finally, the protein bands were imaged with a fluorochemical imaging system (ProteinSimple). The primary antibodies used were against enterovirus 71 3 C (genTex GTX132357), hnRNP A1 (CST 4296), PDIA6 (Proteintech 18233-1-AP), MVP (Univ abs136252), APAF-1 (Proteintech 29022-1-AP), caspase-3 (CST 9662), HNRNPH1 (Proteintech 14774-1-AP), FUS (Proteintech 11570-1-AP), HNRNPU (Proteintech 14599-1-AP), cleaved caspase-3 (CST 9661) for Fig. 7c and Supplementary Fig. 12, cleaved caspase-3 (Affinity, AF7022) for Fig. 7d, and tubulin (Abcam ab59680). Anti-alpha tubulin served as a loading control.

## Immunoprecipitation and mass spectrometry
To identify proteins that interact with the 3 C protease, lysates from 10^7 U87 MG cells transfected with the 3C-FLAG plasmid were

prepared in IP lysis buffer (Beyotime, P0013) supplemented with 1 mM PMSF (Thermo Scientific). Experimental Group: U87-3C-IP cells (cells transfected with the 3C-FLAG plasmid, $n = 3$ biological replicates). Control Group: U87-3C-IgG cells (cells transfected with the 3C-FLAG plasmid and immunoprecipitated with nonspecific IgG, $n = 3$ biological replicates). The lysates were sonicated for 30 s at 20% amplitude, incubated on ice for 1 h, and cleared by centrifugation at 12,000 × g for 10 min at 4 °C. The supernatants were incubated with 2 µg of anti-FLAG antibody and 20 µL of protein A/G agarose beads (Yeasen, 36403ES08) at 4 °C for 3 h. The beads were washed three times with lysis buffer and eluted with 2× SDS loading buffer for immunoblotting.

For mass spectrometry, proteins were separated by SDS–PAGE, and a gel slice ($n = 1$ technical replicate) was excised and processed for in-gel digestion with trypsin (Promega, V5111). Peptides were extracted, desalted with C18 cartridges (Thermo Scientific), and analyzed via LC-MS/MS on a Q Exactive HF–X mass spectrometer (Thermo Scientific) coupled with an EASY–nLC 1200 system (Thermo Scientific). Peptides were separated on a 75 µm × 25 cm C18 column (2 µm, 100 Å) with a 60-min gradient of 0.1% formic acid (mobile phase A) and 0.1% formic acid in 80% acetonitrile (mobile phase B) at a flow rate of 300 nL/min. The mass spectrometer was operated in data-dependent acquisition (DDA) mode, with each cycle including one MS scan (resolution = 60,000, AGC target = $3 \times 10^6$, maximum injection time = 20 ms, scan range = 350–1800 m/z) and 25 MS/MS scans (resolution = 15,000, AGC target = $2 \times 10^5$, maximum injection time = 50 ms). The HCD collision energy was set to 28%, with a quadrupole isolation window of 1.6 Da and a dynamic exclusion time of 35 s. Raw data were processed using MaxQuant[39] (v1.6.6) with the Andromeda search engine against the UniProt Human proteome database (release 2022-03-29). The search parameters included: Enzyme: Trypsin/P with up to two missed cleavages; Fixed modification: Carbamidomethylation (C); Variable modifications: Oxidation (M) and N-terminal acetylation; Mass tolerance: Precursor mass tolerance of 10 ppm and fragment ion mass tolerance of 0.02 Da; False discovery rate (FDR): 1% at both the peptide and protein levels; Protein identification: Proteins were required to have at least one unique peptide; PTM localization: Post-translational modifications (PTMs) were assessed using a PTM score threshold of >0.75.

## Immunohistochemistry and immunofluorescence (IF)

The tumor tissues were collected from the tumor-bearing mice 24 hours after the first administration. The tissues were fixed in 4% PFA overnight at 4 °C and then transferred to 30% sucrose for cryoprotection. The tissues were subsequently embedded in TissueTek O.C.T. (Sakura) and sectioned in the coronal plane at 40 µm using a sliding microtome (AO 860, American Optical). For immunohistochemistry, coronal sections were treated with primary antibodies against Ki67 (ABclonal A20018) and rabbit mAbs against CD3E (ABclonal A19017), CD4 (Abcam ab183685), and CD8A (ABclonal A23081), labeled with horseradish peroxidase (HRP)-labeled antibody (DAKO); and then treated with a diaminobenzidine (DAB) substrate kit (DAKO) for visualization using a slice scanner (KF-PRO-120, KFBIO, China). For immunofluorescence staining, the sections were fixed in 4% formaldehyde for 20 min and washed in PBS. Then, the tumor sections were treated with 0.25% Triton X-100 for 20 min at room temperature. The sections were incubated with primary antibodies against hnRNPA1 (CST 4296) and enterovirus 71 3C (genTex GTX630191) overnight at 4 °C. Afterward, the sections were incubated for 1 hour at room temperature with the corresponding secondary antibodies: goat anti-rabbit Alexa Fluor 488 (Thermo Fisher A32731) and donkey anti-mouse Alexa Fluor 647 (Thermo Fisher A32787). Nuclear counterstaining was performed with DAPI. Images were taken using a confocal laser scanning microscope (Olympus, FV1100).

## Flow cytometry analysis

Resected tumor specimens were processed within 2 hours under sterile conditions. Mechanical dissociation followed by enzymatic digestion in RPMI 1640 medium yielded single-cell suspensions, which were sequentially filtered through 70 µm and 40 µm strainers. Following red blood cell lysis, cells were washed and resuspended in PBS supplemented with 2% FBS. For staining, cells were incubated with a panel of antibodies, including FITC-conjugated anti-mouse CD3ε (BioLegend 100305), PE-conjugated anti-mouse CD8a (BioLegend 100707), FITC-conjugated anti-mouse CD4 (BioLegend 100509), Alexa Fluor® 647-conjugated anti-mouse CD25 (BioLegend 102019), recombinant APC-conjugated granzyme B (BioLegend 372203), and PE-conjugated FOXP3 (BioLegend 320007), on ice for 30 minutes in PBS containing 2% FBS. Flow cytometric analysis was conducted using an LSRFortessa SORP flow cytometer (BD Biosciences, Franklin Lakes, NJ), with data analysis performed using FlowJo software (Ashland, OR).

## Construction of hnRNPA knockout cells

The HNRNPA1 locus was targeted using CRISPR-Cas9 ribonucleoprotein complexes (AisenGene Bioscience) containing hSpCas9 and chimeric sgRNAs. Two guides specific to exons 2-11 were designed using the CRISPR Design Tool (http://crispr.mit.edu): sgRNA1: AAGAGCGATTAGTCCCATTGTGG, sgRNA2: AGGTGGAACCCTAACTATTGAGG. Electroporation was performed using the Neon system (Thermo Fisher), followed by single-cell cloning in 96-well plates. Genomic DNA was purified using the Quick-DNA Miniprep kit (Zymo Research), and the target regions were amplified by PCR with the following primers: F-5′-AATCAGAGCTGTTCCAGGGC-3′; R-5′-TCCTCCTGGTTCTGACTGCT-3′, using Vazyme 2×Taq Master Mix (Dye Plus; Vazyme, P112). Plasmids from 8–10 single colonies were isolated and sequenced using Sanger sequencing (GENEWIZ, China). Clones with mutations in both alleles were selected for further studies. The generated cell lines, including gene (+/+), gene (+/−), gene (+/−A), gene (+/−B), and gene (−/−) cells, were maintained under conditions identical to those of the parental cells.

## FACS analysis

To profile immune cell composition in spleen and blood, samples were collected from treated and control mice post 14-day intervention and immediately maintained in ice-cold PBS. Tissue dissociation was performed by mechanical disruption through a 6-well plate-mounted strainer, followed by homogenization with sterile pestles in cell staining buffer. Cell suspensions were centrifuged (350 × g, 5 min, 4 °C) and erythrocytes were lysed by 10-minute room temperature incubation with RBC lysis buffer. After washing with staining buffer, cells were labeled with: CD3 (BioLegend, 100203), CD19 (BioLegend, 115507), or NK1.1 (BioLegend, 108709) and incubated in the dark for 30 minutes on ice. Following staining, the cells were washed with cell staining buffer, resuspended in PBS, and passed through a 40 µm filter for flow cytometry analysis using an Agilent NovoCyte flow cytometer.

## Statistics and reproducibility

Statistical comparisons were conducted using parametric tests, specifically employing two-tailed Student's t-tests for analyses involving two groups and one-way ANOVA for assessments with multiple groups. Analyses were conducted in GraphPad Prism 8.0 and Microsoft Excel. Data are presented as standard deviation (S.D.) or standard error of the mean (S.E.M.), as specified in the main text. The threshold for statistical significance was set at $^*P < 0.05$, $^{**}P < 0.01$, $^{***}P < 0.001$, and $^{****}P < 0.0001$, with exact P-values provided in source data files. The sample sizes for each experiment are detailed in the corresponding figure legends. Experimental design included random allocation of samples and blinding of investigators to sample collection and data analysis. All experiments were independently replicated at least three

times to ensure reproducibility. No specific statistical method was employed to predetermine sample size.

## Reporting summary

Further information on research design is available in the Nature Portfolio Reporting Summary linked to this article.

## Data availability

Mass spectrometry data has been deposited to the ProteomeXchange Consortium via the iProX repository (https://www.iprox.cn/page/PDV0141.html) with the data set identifier iProX ID (IPX0011224001). All other data supporting this work are included in this article, Supplementary Information or source data files. Source data are provided with this paper.

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

## Acknowledgements

This work was supported by grants from the National Science and Technology Major Projects for "Major New Drugs Innovation and Development", China (2018ZX09711003, to W.Z.).

## Author contributions

X.Y. performed the experiments, drafted the manuscript, analyzed the data and prepared the figures. W.L. and S.Y. carried out the experiments. Z.W. participated in the animal model therapeutic studies. J.Y. and W.Z. conducted the safety studies, including liver enzyme elevation and IHC analysis. H.T. and S.L. performed the tail vein injection assays and part of the WB assay; X.G. and Q.D. carried out the RT–PCR; W.Z., Y.L., X.Y.., C.L., J.L., and S.R. contributed to sample preparation; and P.W. and Y.S. carried out orthotopic HCC implantation and imaging. Y.L., Z.L., and J.Y. contributed to the bioinformatics analysis and plasmid construction. Z.C. contributed to the analytical support. R.C. conceived the idea, designed the experiments and prepared the manuscript, S.L. guided the strategy design, and W.Z. designed the experiments and supervised the study.

## Competing interests

The authors declare no competing interests.
