## [Transparent Peer Review file · Nature Communications]

Gene Therapy with Enterovirus 3C Protease: A Promising Strategy for Various Solid Tumors

Corresponding Author: Professor Wu Zhong

Version 0:

Reviewer comments:

Reviewer #1

(Remarks to the Author)

In this manuscript by Yang and colleagues, the authors aim to develop a novel method of targeted protein degradation. They use enterovirus 3c protease delivered in lipid nanoparticles. They used glioblastoma cell lines and others to show that transfection of 3C-mRNA reduces proliferation in a dose dependent manner. They use a chemical 3C protease inhibitor and the C147S mutant as controls. While it is clear that 3C-LNP affects cell proliferation and viability in vivo and in vitro, without provoking a T cell response and also showing safety data, the major question is whether this is related to broad cleavage effects or specific to just hnRNP A1 as the authors suggest. This question needs to be answered before this report is ready for publication.

- The evidence provided that in U87 MG and U138 MG cells (Fig7) shows that cleaved caspase levels are increased only when 3C-mRNA is transfected in U87 MG and U138MG cells (7D). However, levels of its target hnRNPA1 are only modestly reduced in U138 MG and not at all in U87. A better control would be to knock out hnRNPA1 and determine if cleaved caspase levels change. This would determine the specificity of 3C-mRNA directed cleavage on hnRNPA1.
- The authors chose to test hnRNPA1 because its expression level in GBM tumors was greater than normal. They used Gene Expression Profiling Interactive Analysis as a resource but this gives gene expression data, not protein expression. The analysis shown in Extended Data Fig 5D should show protein levels from a database. In addition, the current figure is not completely labeled, which is the normal tissue? Finally, the difference between normal and GBM appears to be significant for PDIA6 and MVP. These were not affected by 3C-mRNA targeting as shown in Fig 7 so choosing hnRNPA1 is reasonable but were other targets considered? A full list of significantly different proteins should be provided and rationale given for why one or more selected for determining 3C-mRNA target specificity. For example, the authors list 18 proteins with cleavage sites. What determines 3C-mRNA activity if it is not the presence of a cleavage site? This is critical to determining the applicability of this approach more broadly, which is a stated goal of this report.
- The authors show biochemical data in GBM cell lines but show functional data in and HCC and breast tumor model. Is hnRNPA1 also the target in these model systems?

Reviewer #2

(Remarks to the Author)

The manuscript by Yang et al., "A targeted protein degradation strategy based on viral protease for cancer therapy", reports on a novel strategy based on delivering the enterovirus 3C protease to cancer cells and tumors to induce the elimination of malignant cells.

The authors elegantly demonstrate that transfection of an optimized mRNA construct encoding for the 3C protease, but not a proteolytically defective 3C or cells treated with a 3C inhibitor, can result in cancer cell killing. Interestingly, normal cells (HUVEC) remain resistant to cell death by this approach. The authors then show that expression of the 3C protease within cancer cells suppresses proliferation, migration, invasion, and augment apoptosis. Remarkably, when administered to an intracranial model of brain cancer via LNP, the 3C mRNA approach impressively reduced tumor growth and prolonged survival. The authors then confirmed these effects in orthotopic models of breast cancer and hepatocellular carcinoma. The

authors then report on the safety of the approach in mice and interestingly also present results suggesting that an anti-tumor immune response is not involved in the tumor regression observed by their 3C therapy. Finally, the authors propose that the main anti-cancer target of the 3C protease is hnRNP A1.

This is an interesting manuscript describing a proteolytic approach to degrade proteins within cancer cells and tumors and induce cell death via the delivery and expression of a viral protease, in this case the 3C protease from enterovirus. While the results are clearly presented and conclusive, there are several aspects that the authors should address particularly in regard to the mechanism of action and effectiveness in immunocompetent syngeneic models:

The authors describe their approach as a TPD therapeutics. There are dozens of variations on that TPD theme, such as PROTAC LYTAC, AUTAC, bioPROTAC, but they all adhere to the principle of proximity-induced reactions of labelling or ligation by either small-molecules or fusions proteins that artificially bring 2 proteins into contact for proximity-induced effects. This LNP-based delivery of the enterovirus 3C protease in that sense may not be considered a targeted protein degradation approach? Rather, this approach is more a gene therapy approach, with the delivery of an exogenous protease to induce the intracellular cleavage of proteins, much akin to delivering caspases or granzymes or toxins like ricin (which is a ribosomal nuclease in the later case).

While the authors propose that hnRNP A1 is the main target of the 3C protease, this protease can degrade multiple targets as it has been previously reported, and likely the anti-cancer effect of the 3C protease is multi-pronged. To conclude on the hnRNP A1 link, the authors should at least directly prove the point that hnRNP1A is the key target of 3C by co-transfecting cancer cells with a mutant hnRNPA1 that is resistant to cleavage by 3C and show that the mutant can either rescue the 3C-induced killing, and if not, then point to additional important mediator(s) of the effects that has not been elucidated here. Are hnRNP A1 depleted cancer cells also resistant to the 3C approach?

It is interesting that normal cells are not affected by this therapeutic approach. Can the author fully demonstrate that the expression levels of the 3C protease in normal cells is equivalent to those in cancer cells? The testing of other normal cells, other than just HUVEC, would strengthen these results. In addition, demonstrating the effect in both human and mouse (normal and cancer) cells would be important. The safety evaluation in mice presented in the manuscript are conclusive, but the authors should demonstrate that mouse hnRNP A1 is or is not a substrate for the 3C protease, as they have shown for the human counterpart. If mouse hnRNPA1 (or other important/critical targets of the 3C protease) is/are not cleaved in mouse cells or in vivo, then that may account for the good safety profile of 3C-LNP that they see and for the differential killing of human tumor xenograft in mice. The authors do show that GL261 mouse glioblastoma tumors appear affected by 3C administration in a subcutaneous model, although those tumors in the experiment presented are shown to be quite small in nature and kept for only 12 days. Additional syngeneic models would significantly add to the manuscript and help further confirm, or not, the involvement of the immune system in the response observed. Other experiments to determine if T, B or NK cell depletion rescue the therapeutic effect of the 3C approach could also be considered.

The AG7088 could be better explained in the manuscript. In Figure 7, interestingly when the authors use AG7088, the 3C protease band shown in Western blotting appears higher than when no inhibitor is used, for both U87 and U138 transfected cells. Can the authors comment on this, is this how the inhibitor somehow blocks the proteolytic action of 3C?

Reviewer #3

(Remarks to the Author)

Yang, Zhong et al. have developed a potential therapeutic mechanism in which an mRNA expressing a viral protease is packaged in a lipid nanoparticle and delivered to tumor cells for anti-cancer effects. The protease used, 3C protease, exhibits a differential effect on cancer cells vs normal cells in vitro, and appears to support a therapeutic index when delivered in vivo. The authors correlate the anti-cancer activity to downregulation of the protein hnRNPA1. The observation that there may be a therapeutic index associated with this treatment strategy is interesting and warrants further investigation.

The approach appears to have a low technical barrier for further preclinical investigation, however the extent to which delivery and tissue penetration will prove to be limiting for this approach is not fully derisked in this work. mRNA vaccines don't need extensive tissue exposure and cellular coverage, and can get away with transfection of a cellular sub-population, which is not likely to be true in this approach which aims to achieve tumor-intrinsic anti-tumor activity. However, advantages of the approach include the benefit of the catalytic mechanism, which may mean that low levels of protease expression may still have potent effects.

However, the observation of in vitro and in vivo anti-tumor effects is not strongly connected in this work to the proteins observed to be downregulated upon treatment. For example, additional experiments would be required to robustly connect hnRNPA1 to the observed anti-tumor phenotype. hnRNPA1 has been previously characterized as a potential anti-cancer target, somewhat reducing the novelty, however, this target does have druggability challenges that may be addressed by the strategy described in this work. Overall, the authors describe an interesting phenomenon that is worthy of follow-up, however limited mechanistic work and the absence of key controls leaves many open questions, and so I do not recommend the work for publication in Nature Communications in its current form.

Issues

Paragraph one of the introduction: The authors describe the limitations of LYTACs and PROTACs. Clinical TPD efforts are still dominated by molecular glues, to which many of the statements do not apply. Please broaden or qualify the statements regarding TPD to include glues and add an appropriate review.

Figure 4(h) should be Figure 4(f) in the legend.

HUVEC cells do not exhibit anti-proliferative activities, neither do mouse tissues exhibit signs of toxicity during in vivo dosing. These data provide key evidence that dosing of the 3C mRNA may have a therapeutic index. However, 3C protein expression is confirmed only in a limited set of cell lines. It should be confirmed that 3C protease is expressed to equivalent levels in HUVECs and mouse cells, at relevant concentrations, to demonstrate that the differential effects are due to differential sensitivity of tumor vs non-tumor, and not due to technical artifacts or species specific effects. This understanding of safety/therapeutic interest could also be supported in vivo with the pharmacodynamic studies showing that mouse tissues express 3C protease/ down regulate hnRNPA1 as described below.

Similarly, the expression of the C147S mutant 3C protein should be confirmed in lines where it is compared to WT, to ensure that the mutation isn't disrupting expression.

There is limited work linking the in vivo activity of the nanoparticle formulation to the in vitro characterization work. This work would be much more robust with a clear demonstration of in vivo pharmacodynamics. This could include:

- Demonstration of broad/extensive expression of the 3C protease in tumor and adjacent healthy tissues following administration of doses relevant to the in vivo activity and / or:
- Demonstration of down regulation of hnRNPA1 following administration of doses relevant to the in vivo activity

The authors correlate binding and down regulation of hnRNPA1 and a small number of other proteins with the anti-tumor effects of 3C protease transfection. However, there is limited demonstration of causation, and this is a major limitation of the work, which this reviewer believes is critical to address. In the absence of a proven molecular mechanism, the work is limited to the phenotypic observation of anti-tumor effects following exposure to a 3C protease mRNA.

- This could be directly addressed by mutation of the hypothesized hnRNPA1 3C cleavage site in cell lines that are sensitive to the 3C mRNA, demonstration that mutant hnRNPA1 is protected from degradation, and further demonstration that this protects cells from the antiproliferative effects of the treatment.
- This can also be supported by demonstration that hnRNPA1 mRNA levels are not affected ahead of down regulation of hnRNPA1 protein, to support the proposed mechanism which should be post-translational protein degradation.

Version 1:

Reviewer comments:

Reviewer #1

(Remarks to the Author)

Concerns have been well addressed.

Reviewer #2

(Remarks to the Author)

I am satisfied with the revisions made by the authors on the manuscript. Several key experiments were added, strengthening the conclusion of the results and the interesting findings of the paper.

Reviewer #4

(Remarks to the Author)

The authors have addressed previous concerns

The following are our response to the comments of the reviewers and the changes (in blue) in the manuscripts.

Reviewer #1 (cancer therapy, preclinical model):

In this manuscript by Yang and colleagues, the authors aim to develop a novel method of targeted protein degradation. They use enterovirus 3c protease delivered in lipid nanoparticles. They used glioblastoma cell lines and others to show that transfection of 3C-mRNA reduces proliferation in a dose dependent manner. They use a chemical 3C protease inhibitor and the C147S mutant as controls. While it is clear that 3C-LNP affects cell proliferation and viability in vivo and in vitro, without provoking a T cell response and also showing safety data, the major question is whether this is related to broad cleavage effects or specific to just hnRNP A1 as the authors suggest. This question needs to be answered before this report is ready for publication.

- The evidence provided that in U87 MG and U138 MG cells (Fig7) shows that cleaved caspase levels are increased only when 3C-mRNA is transfected in U87 MG and U138MG cells (7D). However, levels of its target hnRNPA1 are only modestly reduced in U138 MG and not at all in U87. A better control would be to knock out hnRNPA1 and determine if cleaved caspase levels change. This would determine the specificity of 3C-mRNA directed cleavage on hnRNPA1.

Thank you for your insightful comments and suggestions! To address this issue, we utilized CRISPR technology to generate hnRNPA1 knockout (KO) U87 MG cell line. Western blot analysis confirmed that the basal level of Cleaved-caspase-3 in the hnRNPA1 knockout cell line was relatively low and did not decrease further upon 3C expression. Moreover, the cell viability of the hnRNPA1 KO cell line remained unaffected by increasing concentrations of 3C-mRNA. These findings suggest that hnRNPA1 is a major target of 3C. We have included these new data into the revised manuscript as Fig.7d and Fig.7e.

Fig. 7| Ectopic expression of 3C protease decreases the level of hnRNP A1 and induces apoptosis through cleavage of hnRNP A1. (d) Western blot analysis of Cleaved-caspase-3, hnRNPA1, expression and 3C expression in U87 MG-WT and U87 MG-hnRNP A1 KO cells transfected with 3C-mRNA and 3C(C147S)-mRNA. (e) Cell viability assay of U87 MG-WT and U87 MG-hnRNP A1 KO cells transfected with different concentrations of 3C-mRNA for 24 h, respectively.

- The authors chose to test hnRNPA1 because its expression level in GBM tumors was

greater than normal. They used Gene Expression Profiling Interactive Analysis as a resource but this gives gene expression data, not protein expression. The analysis shown in Extended Data Fig 5D should show protein levels from a database. In addition, the current figure is not completely labeled, which is the normal tissue? Finally, the difference between normal and GBM appears to be significant for PDIA6 and MVP. These were not affected by 3C-mRNA targeting as shown in Fig 7 so choosing hnRNPA1 is reasonable but were other targets considered? A full list of significantly different proteins should be provided and rationale given for why one or more selected for determining 3C-mRNA target specificity. For example, the authors list 18 proteins with cleavage sites. What determines 3C-mRNA activity if it is not the presence of a cleavage site? This is critical to determining the applicability of this approach more broadly, which is a stated goal of this report.

(1) Thank you for your valuable suggestions on our manuscript! We have taken additional steps to address the need for protein-level data. While we initially used GEPIA2(Gene Expression Profiling Interactive Analysis), an online bioinformatics tool for analyzing gene expression data from TCGA (The Cancer Genome Atlas) and GTEx (Genotype-Tissue Expression) databases, we recognize its limitations in providing direct protein expression information. Therefore, we turned to The Human Protein Atlas (HPA) to analyze protein expression differences in GBM tumors and employed the WB to further validate it. This finding is significant as it indirectly supports the notion that the 3C primarily targets and downregulates hnRNPA1. We have included these new data into the revised manuscript as Extended Data Fig. 7.

Extended Data Fig. 7| 3C protease does not affect protein expression of hnRNPU, hnRNPH1 and FUS in GBM cells. Western blot analysis of hnRNPU, hnRNPH1, FUS and 3C expression in U87 MG and U138 MG cells transfected with 3C-mRNA and 3C(C147S)-mRNA, and in cells transfected with 3C-mRNA and treated with AG7088.

(2) Thank you for your good suggestion! We have added annotations in the figure legend of Extended Data Fig 6.d.

(3) Thank you for your question regarding the determinants of 3C-mRNA activity beyond the presence of a cleavage site. The activity of the enterovirus 3C protease, delivered via 3C-mRNA, is determined by several key factors:

Amino Acid Sequences Targeted by 3C Protease: The 3C protease has a stringent specificity for certain amino acid sequences, which is pivotal to its activity and therapeutic potential. The 3C protease recognizes and cleaves after the consensus

sequence of glutamine (Q) and glycine (G), known as the QS or QG sequence. This selectivity is crucial for the targeted degradation of specific proteins, such as hnRNPA1, which we have shown to be a major target of 3C protease in our study. The ability of the 3C protease to selectively cleave these sequences is a cornerstone for its activity and is critical for understanding the broader applicability of this approach.

Catalytic Triad: The activity of the 3C protease is intrinsically linked to its catalytic triad, which is composed of cysteine (Cys), histidine (His), and aspartate (Asp). These amino acid residues play a direct role in substrate recognition and catalytic hydrolysis. It is important to note that the C147S mutation, which we mentioned in our study, is based on previous research indicating that such a mutation can inactivate the enzyme's protease activity¹⁴. We constructed the C147S mutation in the 3C-mRNA to further investigate the impact of this change on the enzyme's activity and antitumor effects. Our experiments demonstrated that the C147S mutation abrogates the enzyme's activity, supporting the significance of the catalytic triad for the biological function of the 3C protease.

By understanding these mechanisms, we can better predict the activity of 3C-mRNA and its potential for targeted protein degradation. The catalytic triad's role, along with the 3C protease's stringent specificity for the QS and QG sequences, stands out as a pivotal determinant of its activity and a cornerstone for its therapeutic potential.

14 Weng, K. F., Li, M. L., Hung, C. T. & Shih, S. R. Enterovirus 71 3C protease cleaves a novel target CstF-64 and inhibits cellular polyadenylation. *PLoS Pathog* 5, e1000593, doi:10.1371/journal.ppat.1000593 (2009).

- The authors show biochemical data in GBM cell lines but show functional data in and HCC and breast tumor model. Is hnRNPA1 also the target in these model systems?

Thank you for your valuable suggestion! We have conducted additional WB analyses to examine the protein levels of hnRNPA1 in HCC and MCF7 cell lines following treatment with 3C-mRNA, and have included these data in the revised manuscript's Extended Data Figure 12. The WB data demonstrate that the levels of hnRNPA1 are significantly reduced upon 3C-mRNA treatment in both HCC and MCF7 cells, which is consistent with our observations in GBM cell lines.

Extended Data Fig. 12| The expression levels of hnRNPA1 are significantly reduced upon 3C expression in both Huh7 and MCF7 cells. Western blot analysis of Cleaved-caspase-3, hnRNPA1, and 3C expression in Huh7 and MCF-7 cells transfected with different concentrations 3C-mRNA.

Reviewer #2 (Oncolytic virotherapy):

The manuscript by Yang et al., “A targeted protein degradation strategy based on viral protease for cancer therapy”, reports on a novel strategy based on delivering the enterovirus 3C protease to cancer cells and tumors to induce the elimination of malignant cells.

The authors elegantly demonstrate that transfection of an optimized mRNA construct encoding for the 3C protease, but not a proteolytically defective 3C or cells treated with a 3C inhibitor, can result in cancer cell killing. Interestingly, normal cells (HUVEC) remain resistant to cell death by this approach. The authors then show that expression of the 3C protease within cancer cells suppresses proliferation, migration, invasion, and augment apoptosis. Remarkably, when administered to an intracranial model of brain cancer via LNP, the 3C mRNA approach impressively reduced tumor growth and prolonged survival. The authors then confirmed these effects in orthotopic models of breast cancer and hepatocellular carcinoma. The authors then report on the safety of the approach in mice and interestingly also present results suggesting that an anti-tumor immune response is not involved in the tumor regression observed by their 3C therapy. Finally, the authors propose that the main anti-cancer target of the 3C protease is hnRNP A1.

This is an interesting manuscript describing a proteolytic approach to degrade proteins within cancer cells and tumors and induce cell death via the delivery and expression of a viral protease, in this case the 3C protease from enterovirus. While the results are clearly presented and conclusive, there are several aspects that the authors should address particularly in regard to the mechanism of action and effectiveness in immunocompetent syngeneic models:

The authors describe their approach as a TPD therapeutics. There are dozens of variations on that TPD theme, such as PROTAC, LYTAC, AUTAC, bioPROTAC, but they all adhere to the principle of proximity-induced reactions of labelling or ligation by either small-molecules or fusions proteins that artificially bring 2 proteins into contact for proximity-induced effects. This LNP-based delivery of the enterovirus 3C protease in that sense may not be considered a targeted protein degradation approach? Rather, this approach is more a gene therapy approach, with the delivery of an exogenous protease to induce the intracellular cleavage of proteins, much akin to delivering caspases or granzymes or toxins like ricin (which is a ribosomal nuclease in the later case).

Thank you for your insightful commentary and for highlighting the distinction between targeted protein degradation (TPD) therapeutics and our approach involving the delivery of the enterovirus 3C protease. We have updated the title of our manuscript to better represent our work.

The title changed to “Gene Therapy with Enterovirus 3C Protease: A Promising Strategy for Glioblastoma and Other Tumors”

While the authors propose that hnRNP A1 is the main target of the 3C protease, this protease can degrade multiple targets as it has been previously reported, and likely the anti-cancer effect of the 3C protease is multi-pronged. To conclude on the hnRNP A1 link, the authors should at least directly prove the point that hnRNP1A is the key target of 3C by co-transfecting cancer cells with a mutant hnRNPA1 that is resistant to cleavage by 3C and show that the mutant can either rescue the 3C-induced killing, and if not, then point to additional important mediator(s) of the effects that has not been elucidated here. Are hnRNP A1 depleted cancer cells also resistant to the 3C approach? Thank you for your valuable suggestion! In response to your suggestion, we have generated hnRNPA1 knockout (KO) U87 MG cell line using CRISPR-Cas9 technology. Our results demonstrate that the hnRNPA1 KO cells exhibit reduced sensitivity to the 3C protease compared to wild-type cells, suggesting that hnRNPA1 is indeed a primary target of the 3C protease. Western blot analysis further revealed that the level of Cleaved-caspase-3 in the mutant cell line was comparable to that in the knockout, remaining relatively low and not showing further reduction in response to 3C expression. These findings reinforce the conclusion that hnRNP A1 is a major factor involved in the 3C protease's mechanism. We have included these new data into the revised manuscript as Fig.7d and Fig.7e.

Fig. 7| Ectopic expression of 3C protease decreases the level of hnRNP A1 and induces apoptosis through cleavage of hnRNP A1. (d) Western blot analysis of Cleaved-caspase-3, hnRNPA1, expression and 3C expression in U87 MG-WT and U87 MG-hnRNP A1 KO cells transfected with 3C-mRNA and 3C(C147S)-mRNA. (e) Cell viability assay of U87 MG-WT and U87 MG-hnRNP A1 KO cells transfected with different concentrations of 3C-mRNA for 24 h, respectively.

It is interesting that normal cells are not affected by this therapeutic approach. Can the author fully demonstrate that the expression levels of the 3C protease in normal cells is equivalent to those in cancer cells? The testing of other normal cells, other than just HUVEC, would strengthen these results. In addition, demonstrating the effect in both human and mouse (normal and cancer) cells would be important. The safety evaluation in mice presented in the manuscript are conclusive, but the authors should demonstrate that mouse hnRNP A1 is or is not a substrate for the 3C protease, as they have shown for the human counterpart. If mouse hnRNPA1 (or other important/critical targets of the 3C protease) is/are not cleaved in mouse cells or in vivo, then that may account for the good safety profile of 3C-LNP that they see and for the differential killing of human tumor xenograft in mice.

Thank you for your valuable suggestion! We have now included WB validation in HUVEC, 293T (human renal epithelial cells), Huh7 (hepatocellular carcinoma cells), and MCF-7 (breast cancer cells) to compare the expression levels of the 3C protease, and have included these data in the revised manuscript's Extended Data Figure 11. Our WB results consistently show that the expression levels of the 3C protease in these normal and cancer cell lines are equivalent.

Extended Data Fig. 11| 3C expression in different human normal and cancer cells. Western blot analysis of 3C expression in HUVEC, 293T, MCF-7 and Huh7 cells transfected with 3C-mRNA and 3C(C147S)-mRNA.

For the mouse cell lines, we have included MEF (Mouse Embryonic Fibroblasts), GL261 (mouse glioma cells), and 4T1 (mouse breast cancer cells) in our study, and have included these data in the revised manuscript's Extended Data Figure 13. Our WB results clearly indicate that the expression levels of the 3C protease are comparable among these mouse cell lines. Importantly, we have also observed that the expression of 3C leads to a downregulation of hnRNPA1 levels in all these mouse cell lines. This finding not only shows that the 3C protease is active in the mouse cellular context but also that it affects hnRNPA1, suggesting a conserved mechanism between human and mouse cells.

Extended Data Fig. 13| 3C expression in different mouse normal and cancer cells. Western blot analysis of hnRNPA1 and 3C expression in GL261, 4T1, and MEF cells transfected with 3C-mRNA and 3C(C147S)-mRNA.

The authors do show that GL261 mouse glioblastoma tumors appear affected by 3C administration in a subcutaneous model, although those tumors in the experiment presented are shown to be quite small in nature and kept for only 12 days. Additional syngeneic models would significantly add to the manuscript and help further confirm, or not, the involvement of the immune system in the response observed. Other experiments to determine if T, B or NK cell depletion rescue the therapeutic effect of the 3C approach could also be considered.

Thank you for your valuable comments and suggestions! We have extended our study to include the 4T1 cell line in BALB/c mice, a syngeneic breast cancer model, to further investigate the immune response, and incorporated these data into the revised manuscript as Extended Fig. 3 h, i, j, k, l, m, n.

Also, we conducted T, B, and NK cell depletion experiments in C57 mice to thoroughly investigate the role of the immune system in the therapeutic effect of the 3C approach. Our findings indicate that the depletion of T cells, B cells, or NK cells did not abrogate the therapeutic effect of 3C-LNP, suggesting that the anti-tumor effects are not mediated by the immune system, but through direct tumor cell killing. We have included these new results into the revised manuscript as Extended Fig. 4.

Extended Fig. 3 | The antitumor activity of 3C-LNPs is independent of T cell-mediated antitumor immune response. (a) Timeline of tumor implantation and treatment schedule in the subcutaneous tumor model of C57BL/6 mice. Tumor-bearing mice were subcutaneously injected with PBS or 3C-LNPs (3 μg) every 3 days for a total of four doses. (n=5 mice/group). (b) Tumor growth curves of the tumor-bearing mice after treatment with PBS or 3C-LNPs (n=5 mice/group). The data are presented as the means ± SEMs. Statistical differences were assessed using one-way ANOVA with the Bonferroni multiple comparisons test. (c-e) and (j-l) Representative IHC staining images of tumor tissues and quantification of CD3, CD4, and CD8+ T cells per field. The data are presented as the means ± SEMs. Statistical differences were assessed using multiple t tests. n.s.= not significant. (f) and (m) Quantification of cytotoxic T lymphocytes. (g) and (n) Quantifications of regulatory T lymphocytes. (h) Timeline of tumor

implantation and treatment schedule in the 4T1 mammary fat pad tumor model of Balb/c mice. Mice were subcutaneously injected around the tumor tissues with PBS, or 3C-LNPs (3 μ g) every 3 days for a total of four doses (n=6 mice/group). The data are presented as the means \pm SDs. Statistical differences were assessed using an unpaired t test. n.s.=not significant. * $P < 0.05$, ** $P < 0.01$, *** $P < 0.001$, **** $P < 0.0001$.

Extended Fig. 4 | The antitumor activity of 3C-LNPs is independent of tumor immune response. Flow cytometry was employed to quantify the levels of (a) CD3+ T cells, (c) CD19+ B cells, and (e) NK1.1+ NK cells in spleen and blood of mice subjected to depletion protocols, two weeks after the initiation of weekly intraperitoneal (i.p.) antibody injections. The data are presented as the means \pm SDs. Statistical differences were assessed using two-way ANOVA with the Bonferroni multiple comparisons test. (b), (d) and (f) Tumor growth curves of the tumor-bearing mice after treatment with PBS combined with antibodies, PBS combined with corresponding control IgG, 3C-LNP combined with antibodies, and 3C-LNP combined with corresponding control IgG (n = 6 mice/group). The data are presented as the means \pm SEMs. Statistical differences were assessed using one-way ANOVA with the Bonferroni multiple comparisons test. n.s.=not significant. **** $P < 0.0001$.

The AG7088 could be better explained in the manuscript. In Figure 7, interestingly when the authors use AG7088, the 3C protease band shown in Western blotting appears higher than when no inhibitor is used, for both U87 and U138 transfected cells. Can the authors comment on this, is this how the inhibitor somehow blocks the proteolytic action of 3C?

Thank you for your valuable suggestions! We have carefully considered this observation and would like to provide an explanation based on the available evidence. AG7088, also known as Rupintrivir, is a potent and irreversible inhibitor of 3C protease. The compound has been shown to form a covalent bond with the active site cysteine of the 3C protease, leading to its inactivation. This irreversible inhibition suggests that AG7088 may alter the conformation of the 3C protease, potentially affecting its interaction with other cellular factors or its stability. We also noticed that there could be some minor differences in the protein gel preparation and electrophoresis buffer. These factors might contribute to the observed variation in the band intensity. We have re -

examined our protein gel preparation process and performed several repetitions of the experiments, and we are attaching the original pictures for your reference. This further supports our understanding of the phenomenon and we hope it will address your concerns.

Reviewer #3 (Protein-targeted degradation, cancer therapy):

Yang, Zhong et al. have developed a potential therapeutic mechanism in which an mRNA expressing a viral protease is packaged in a lipid nanoparticle and delivered to tumor cells for anti-cancer effects. The protease used, 3C protease, exhibits a differential effect on cancer cells vs normal cells in vitro, and appears to support a therapeutic index when delivered in vivo. The authors correlate the anti-cancer activity to downregulation of the protein hnRNPA1. The observation that there may be a therapeutic index associated with this treatment strategy is interesting and warrants further investigation.

The approach appears to have a low technical barrier for further preclinical investigation, however the extent to which delivery and tissue penetration will prove to be limiting for this approach is not fully derisked in this work. mRNA vaccines don't need extensive tissue exposure and cellular coverage, and can get away with transfection of a cellular sub-population, which is not likely to be true in this approach which aims to achieve tumor-intrinsic anti-tumor activity. However, advantages of the approach include the benefit of the catalytic mechanism, which may mean that low levels of protease expression may still have potent effects.

However, the observation of in vitro and in vivo anti-tumor effects is not strongly connected in this work to the proteins observed to be downregulated upon treatment. For example, additional experiments would be required to robustly connect hnRNPA1 to the observed anti-tumor phenotype. hnRNPA1 has been previously characterized as

a potential anti-cancer target, somewhat reducing the novelty, however, this target does have druggability challenges that may be addressed by the strategy described in this work. Overall, the authors describe an interesting phenomenon that is worthy of follow-up, however limited mechanistic work and the absence of key controls leaves many open questions, and so I do not recommend the work for publication in Nature Communications in its current form.

Issues

Paragraph one of the introduction: The authors describe the limitations of LYTACs and PROTACs. Clinical TPD efforts are still dominated by molecular glues, to which many of the statements do not apply. Please broaden or qualify the statements regarding TPD to include glues and add an appropriate review.

Thank you for your valuable suggestions on the introduction section of our manuscript. We have made the following revisions to our manuscript:

Title Correction: We have updated the title of our manuscript.

Introduction Revision: We have revised the introduction to emphasize the mRNA gene therapy approach of our study.

Figure 4(h) should be Figure 4(f) in the legend.

Thank you for your kind suggestion! We have corrected it.

HUVEC cells do not exhibit anti-proliferative activities, neither do mouse tissues exhibit signs of toxicity during in vivo dosing. These data provide key evidence that dosing of the 3C mRNA may have a therapeutic index. However, 3C protein expression is confirmed only in a limited set of cell lines. It should be confirmed that 3C protease is expressed to equivalent levels in HUVECs and mouse cells, at relevant concentrations, to demonstrate that the differential effects are due to differential sensitivity of tumor vs non-tumor, and not due to technical artifacts or species specific effects. This understanding of safety/therapeutic interest could also be supported in vivo with the pharmacodynamic studies showing that mouse tissues express 3C protease/down regulate hnRNPA1 as described below.

Similarly, the expression of the C147S mutant 3C protein should be confirmed in lines where it is compared to WT, to ensure that the mutation isn't disrupting expression.

Thank you for your insightful comments and suggestions! We have added the conducted additional experiments to confirm the expression of the 3C protease in HUVECs and mouse cells in the revised manuscript. Importantly, we have also observed that the expression of 3C leads to a downregulation of hnRNPA1 levels in all these mouse cell lines. This finding not only shows that the 3C protease is active in the mouse cellular context but also that it affects hnRNPA1, suggesting a conserved mechanism between human and mouse cells.

Extended Data Fig. 11| 3C expression in different human normal and cancer cells. Western blot analysis of 3C expression in HUVEC, 293T, MCF-7 and Huh7 cells transfected with 3C-mRNA and 3C(C147S)-mRNA.

Extended Data Fig. 13| 3C expression in different mouse normal and cancer cells. Western blot analysis of hnRNPA1 and 3C expression in GL261, 4T1, and MEF cells transfected with 3C-mRNA and 3C(C147S)-mRNA.

There is limited work linking the in vivo activity of the nanoparticle formulation to the in vitro characterization work. This work would be much more robust with a clear demonstration of in vivo pharmacodynamics. This could include:

- Demonstration of broad/extensive expression of the 3C protease in tumor and adjacent healthy tissues following administration of doses relevant to the in vivo activity and / or:
- Demonstration of down regulation of hnRNPA1 following administration of doses relevant to the in vivo activity

Thank you for your constructive suggestions! We have conducted further experiments in a glioblastoma (GBM) orthotopic model. The immunofluorescence results show a clear downregulation of hnRNPA1 in the tumor tissues, which is consistent with the doses relevant to the in vivo activity. Conversely, the mutant 3C expression exhibited a similar pattern to the PBS control group, with no discernible effect on hnRNPA1 levels. We have incorporated these findings into the revised manuscript as Extended Data Fig. 10.

Extended Data Fig. 10 | In vivo validation of 3C protein expression effects on hnRNPA1 levels and apoptosis induction in GBM models. (a) Immunofluorescence staining of tumor and adjacent sections showing the expression of hnRNPA1 (green), 3C protein (red), and cellular nuclei (DAPI, blue). The merge images illustrate co-localization. Scale bars= 20 μm. (b) Quantification of hnRNPA1 expression relative to DAPI. The data are presented as the means ± SD. Statistical differences were assessed using one-way ANOVA. n.s.=not significant. **** $P < 0.0001$.

The authors correlate binding and down regulation of hnRNPA1 and a small number of other proteins with the anti-tumor effects of 3C protease transfection. However, there is limited demonstration of causation, and this is a major limitation of the work, which this reviewer believes is critical to address. In the absence of a proven molecular mechanism, the work is limited to the phenotypic observation of anti-tumor effects following exposure to a 3C protease mRNA.

- This could be directly addressed by mutation of the hypothesized hnRNPA1 3C cleavage site in cell lines that are sensitive to the 3C mRNA, demonstration that mutant hnRNPA1 is protected from degradation, and further demonstration that this protects cells from the antiproliferative effects of the treatment.

- This can also be supported by demonstration that hnRNPA1 mRNA levels are not affected ahead of down regulation of hnRNPA1 protein, to support the proposed mechanism which should be post-translational protein degradation.

Thank you for your insightful comments! We have generated hnRNPA1 knockout (KO) U87 MG cell line using CRISPR-Cas9 technology. Our results demonstrate that the hnRNPA1 KO cells do not exhibit the same sensitivity to 3C protease as the wild-type cells, suggesting that hnRNPA1 is indeed a primary target of 3C protease. Western blot analysis further revealed that the level of Cleaved-caspase-3 in the mutant cell line was comparable to that in the knockout, remaining relatively low and not showing further reduction in response to 3C expression. These findings reinforce the conclusion that hnRNPA1 is a major factor involved in the 3C protease's mechanism. We have included these new data into the revised manuscript as Fig.7d and Fig.7e.

Regarding the construction of hnRNPA1 mutants, we encountered a complexity due to the presence of multiple QG and QS sequences within the hnRNPA1 protein. To precisely locate the cleavage site, more elaborate molecular mechanism experiments are required, which involves performing mutations at several potential sites to identify the specific cleavage site. We acknowledge this as an important aspect and will continue to validate it in our future work.

Fig. 7| Ectopic expression of 3C protease decreases the level of hnRNP A1 and induces apoptosis through cleavage of hnRNP A1. (d) Western blot analysis of Cleaved-caspase-3, hnRNPA1, expression and 3C expression in U87 MG-WT and U87 MG-hnRNP A1 KO cells transfected with 3C-mRNA and 3C(C147S)-mRNA. (e) Cell viability assay of U87 MG-WT and U87 MG-hnRNP A1 KO cells transfected with different concentrations of 3C-mRNA for 24 h, respectively.

We also conducted relative quantitative PCR (qPCR) experiments to evaluate the mRNA levels of hnRNPA1 following treatment with both 3C-mRNA and the 3C(C147S)-mRNA in U87 MG cells and U138 MG cells. Our findings reveal that there are no significant differences in hnRNPA1 mRNA levels across the three groups, which underscores that the observed downregulation of hnRNPA1 is post-translational and not due to changes at the transcriptional level. These results are now included in the revised manuscript.

Extended Data Fig. 9| mRNA expression levels of hnRNPA1 in U87 MG and U138 MG cells. mRNA expression levels of hnRNPA1 in U87 MG and U138 MG cells following transfection with control (lipo2000), 3C-mRNA (3.5 μ g), and 3C(C147S)-mRNA (3.5 μ g). The mRNA relative expression was normalized to GAPDH and is presented as the mean \pm SD. Statistical differences were assessed using one-way ANOVA. n.s.=not significant.